# Thematic Analysis of Indonesian Physics Education Research Literature Using Machine Learning

**Purwoko Haryadi Santoso** [1,2,*] , **Edi Istiyono** [1,3] , **Haryanto** [1] **and Wahyu Hidayatulloh** [3]

1 Graduate School of Educational Research and Evaluation, Universitas Negeri Yogyakarta, Sleman 55281, Indonesia
2 Department of Physics Education, Universitas Sulawesi Barat, Majene 91412, Indonesia
3 Department of Physics Education, Universitas Negeri Yogyakarta, Sleman 55281, Indonesia
* Correspondence: purwokoharyadi.2021@student.uny.ac.id or purwokoharyadisantoso@unsulbar.ac.id

**Abstract:** Abundant physics education research (PER) literature has been disseminated through academic publications. Over the years, the growing body of literature challenges Indonesian PER scholars to understand how the research community has progressed and possible future work that should be encouraged. Nevertheless, the previous traditional method of thematic analysis possesses limitations when the amount of PER literature exponentially increases. In order to deal with this plethora of publications, one of the machine learning (ML) algorithms from natural language processing (NLP) studies was employed in this paper to automate a thematic analysis of Indonesian PER literature that still needs to be explored within the community. One of the well-known NLP algorithms, latent Dirichlet allocation (LDA), was used in this study to extract Indonesian PER topics and their evolution between 2014 and 2021. A total of 852 papers (~4 to 8 pages each) were collectively downloaded from five international conference proceedings organized, peer reviewed, and published by Indonesian PER researchers. Before their topics were modeled through the LDA algorithm, our data corpus was preprocessed through several common procedures of established NLP studies. The findings revealed that LDA had thematically quantified Indonesian PER topics and described their distinct development over a certain period. The identified topics from this study recommended that the Indonesian PER community establish robust development in eight distinct topics to the present. Here, we commenced with an initial interest focusing on research on physics laboratories and followed the research-based instruction in late 2015. For the past few years, the Indonesian PER scholars have mostly studied 21st century skills which have given way to a focus on developing relevant educational technologies and promoting the interdisciplinary aspects of physics education. We suggest an open room for Indonesian PER scholars to address the qualitative aspects of physics teaching and learning that is still scant within the literature.

**Keywords:** thematic analysis; Indonesia; physics education research; machine learning





## 1. Introduction

Several decades of physics education research (PER) have established an enormous body of literature related to physics teaching and learning. Outside the context of the Indonesian PER community, many thousands of PER articles have been published in several high impact journals, such as *The Physics Teacher* (TPT), *The American Journal of Physics* (AJP), and *Physical Review Physics Education Research* (PRPER) (previously announced as *Physical Review Special Topics Physics Education Research*) since 1933, 1963, and 2005, respectively. We term it as "outside" since representatives of Indonesian PER scholars within these journals are still scant. It must be considered that unique findings from the Indonesian environment are still missing based on these references.

Rare representation of Indonesian PER scholars covered in these journals cannot be translated as the absence of PER development within the Indonesian community. Since 2014

to date, several international conferences in the area of PER have been annually organized by several Indonesian teacher education institutions (TEIs). The five oldest conferences on science, technology, engineering, and mathematics (STEM) education have included the topic of physics education research (PER) for publication. They comprise the International Conference on Research, Implementation, & Education of Mathematics and Science (ICRIEMS, since 2014) [1] and the International Seminar on Science Education (ISSE, since 2015) organized by Universitas Negeri Yogyakarta (UNY) [2], the International Conference on Mathematics & Science Education (ICMSE, since 2014) organized by Universitas Negeri Semarang (UNNES) [3], the International Conference on Mathematics and Science Education (ICMScE, since 2016) organized by Universitas Pendidikan Indonesia (UPI) [4], and the International Conference on Mathematics and Science Education (ICoMSE, since 2017) organized by Universitas Negeri Malang (UM) [5]. These selected international conferences have substantially contributed to our research insights into the Indonesian PER field. Otherwise, peer-reviewed journals were only published nationally during the same timeframe and a smaller number of publications than the aforementioned conferences. Furthermore, they have attracted PER scholars of various backgrounds from novice researchers (graduate students) to PER experts (senior scholars and professors) funded through research grants from the Indonesian government. Mostly, the authors have been affiliated with several Indonesian institutions and a few with neighboring countries, particularly from Southeast Asia region.

Essentially, this volume of publications provides a convincing challenge for PER scholars to understand how the research community has progressed and possible future work that should be emphasized. Nevertheless, it can be troublesome to synthesize whole articles published within a large number of publications. Most researchers tend to review only the most relevant research articles for their work. There is always a possibility that they have neglected some academic resources within the collection of literature. We believe that it is imperative to have insight into PER researchers to further their understanding of PER. These cases are more complicated for novice researchers, who should exhaustively review the extensive development of the field [6]. Consequently, they are usually more dependent on the given suggestions either provided by communities, research groups, or indexing databases like Google Scholar [7].

On the other hand, the number of works could inevitably be perceived as the Indonesian PER field having currently developed to a phase of maintaining its research merit of theoretical and methodological practice through their continued existence for a certain time. Hence, this body of literature is valuable in explaining the characteristics of the Indonesian PER field and its development of topics over time. To synthesize a comprehensive story of PER topics outside the Indonesian PER field, one must consult the previously ambitious work that has been disseminated by McDermott & Redish [8], Docktor and Mestre [9], Meltzer and Otero [10], Odden et al. [11], and Yun [12]. These great works admittedly have guided the PER community in several parts of the world, including the Indonesian PER scholars. Nevertheless, as clearly mentioned before, the representation of Indonesian scholars covered by these disseminations is still limited to best capture the Indonesian PER findings. It might be less appropriate to understand the characteristics and development of the Indonesian PER topics if we merely considered those resources without sufficient involvement of Indonesian PER scholars. Therefore, our current paper extends the intention of previous works to analyze the Indonesian PER field through the methodology of thematic analysis. We believe that addressing this issue should be considered a potential contribution to enrich the merit of previous references. In this paper, we studied 852 proceeding papers organized, peer reviewed, and published by the Indonesian PER community that are unknown from previous works. To the best of our knowledge, Indonesian PER researchers have not yet performed work to analyze their research literature using the similar method performed by our study. Instead, a recent study by Hartono, et al. [13] (Indonesian author) investigated a data corpus outside the context of Indonesia.

Although Indonesian PER research is still scant with regard to performing a thematic analysis, we must admit that other aims related to Indonesian PER have made several efforts in this area, particularly through the conventional method of content analysis on science education [14], scientific literacy [15], teacher education [16], and learning media [17]. However, one may argue that conducting a thematic analysis through traditionally reading and summarizing the vast amount of literature is inefficient. For instance, a recent study on science education research reported by Faisal et al. [14] even argued that performing this sort of analysis on a large number of articles was "tricky", as mentioned in their introduction of a paper about mapping the research trends in Indonesian science education research. Hence, they considered that a content analysis approach on the keywords of proposed titles of research grants was more doable to simplify their study. In their conclusion, Faisal et al. [14] conceded that the selection of this method of keyword-based analysis was problematic in representing the final state of research dissemination. The initial title of the research grant was more likely to be improved after the work had been finished, and either theoretical or methodological considerations may have made it possible for some improvements to occur. Publication of their work might have slightly evolved from the proposed title of the initial announcement of the research grant.

Furthermore, the traditional method of content analysis fails to satisfy the principle of the distributional hypothesis of topics established by the linguistic field [18]. The nature of research topics should demonstrate a mixture of words instead of a single keyword [19]. Consequently, the principle of thematic analysis needs several words to represent a literature topic. Therefore, the mixed membership idea and the distributional hypothesis of topics should be consulted to shed more light on the analysis of literature topics. For this reason, a new more efficient and significant method of thematic analysis should be approached to complete our understanding about the literature topics.

Over the past few years, machine learning (ML) has rapidly become a powerful tool to respond to the growing size of data emerging in the digital era. Textual data is one of the data structures studied within this field. Natural language processing (NLP) is one of the ML studies concerned with sets of texts. NLP proposes a method of thematic analysis to extract our understanding of textual data based on a large collection of literature. Recent studies by Odden et al. [20] have performed this sort of analysis towards *Physics Education Research Conference* (PERC) proceedings [11] and Yun [12] towards the *American Journal of Physics* (AJP) and the *Physical Review Physics Education Research* (PRPER). In this paper, we extend these efforts to analyze Indonesian PER literature using the NLP algorithm. We have performed one of the popular NLP algorithms, latent Dirichlet allocation (LDA) [21,22], to automate a thematic analysis of Indonesian PER literature selected from the five longest running international conference proceedings organized, peer reviewed, and published by the Indonesian PER community between 2014 and 2021. Throughout the LDA topic modeling, we have extracted eight characteristics of Indonesian PER topics and how those topics have been developed within the field over a certain period.

Our contribution to this paper is intended to demonstrate the LDA algorithm in Indonesian PER literature. It has the potential ability to help PER scholars extract valuable information from the vast number of Indonesian PER literature. It inevitably could extract the discovered Indonesian PER topics based on the nature of topics and their associated rise and fall within the field over a certain publication time frame. This study then will be guided by the following two research questions:

RQ1. Using LDA topic modeling towards the five Indonesian PER publications, what are the topic characteristics studied between 2014 and 2021?

RQ2. How has the development of these topics occurred between 2014 and 2021?

The extracted Indonesian PER topics from this study are dedicated to enriching our knowledge about research activities that have been attempted and suggesting areas of further investigation. The demonstration of the promising analytic approach would be our trigger to the wider academic publications within the Indonesian PER community.

## 2. Theoretical Review

Thematic analysis is one type of literature research methodology used in collecting, reviewing, summarizing, and synthesizing previous studies about specific domains [23]. Naturally, thematic analysis is established in the climate of qualitative inquiry. It is constructed, and has similarities, with other systematic procedures of qualitative analysis as demonstrated by grounded theory, narrative analysis, interpretative phenomenological analysis, and content analysis in analyzing personal experience about phenomena [24,25]. The early research practices of a literature review using thematic analysis is undertaken through the constructivist paradigm that the researcher is the main actor in the data collection and analysis [25,26]. Therefore, human-based analysis plays a vital role to conduct the time-consuming literature review using traditional thematic analysis [27]. As briefly discussed in the introduction above, this way encounters serious disadvantages when the number of pieces of literature significantly increases [14]. It also has the potential to make unstable findings, particularly those that are undertaken by novice researchers [28]. Snyder [23] even argues that traditional thematic analysis often produces a lack of thoroughness and rigor-specific methodology. Therefore, several researchers recommend the enhancement of this conventional way to strengthen its robustness for literature reviews. They propose automation technology [29], computational toolkit [30], as well as using machine learning (ML) technology, as demonstrated by the current paper.

Natural language processing (NLP) is the subfield of ML studies that performs topic modeling or text analysis from a set of documents. Broadly speaking, there are two types of ML models, namely supervised and unsupervised algorithms. The supervised ML model specifies a predetermined set of labels in fitting, predicting, or classifying the trained subset of data. Conversely, unsupervised ML models do not specify the desired labels in advance. Accordingly, in an unsupervised NLP model, we do not have a predetermined set of results before processing the text analysis. They rather intend to extract latent entities from a set of documents without knowing the desired results previously. Thereafter, this technique naturally may be troublesome for the interpretation of extracted topics due to the absence of predetermined labels. However, this disadvantage simultaneously often occurred in common text analysis studies [31]. Therefore, NLP researchers must evaluate their interpretations of the extracted topics through several procedures of evaluation metrics explained in the subsequent methodological section of this paper.

Latent Dirichlet allocation (LDA) is a popular unsupervised NLP algorithm that has been commonly used to extract the essence of diverse literature. Even though this text analysis technique has been disadvantaged with some simplifications as explained above, several fields have employed this method persuasively. Since Blei et al. [21] published their LDA algorithm in 2003, LDA has been employed for several purposes such as analyzing customers' opinions in agricultural companies [32], commercial reviews [33], political issues [34], and topics in online news portals [35]. Additionally, LDA also has been implemented in the educational environment to analyze informatics engineering studies [36], project reports [37], undergraduate theses [38], scientific papers [39], and online educational resources [40,41]. Therefore, these numerous LDA implementations offer a promising tool in many fields, including physics education research (PER). Recently, the LDA method has been implemented for the subject of PER [11] in the analysis of large numbers of individual papers from physics education research conference proceedings (PERC) [11]. However, this previous attempt was intended to cover outside the Indonesian context. Thus, it can be less representative for grasping the full knowledge about the development of Indonesian PER studies. To enrich the insight into Indonesian PER development, we believe that analyzing the Indonesian PER literature using the LDA algorithm could be the potential contribution of our paper. Thus, it should be worthwhile since there is little known about how our Indonesian PER community has been established and where we are going further to develop our community.

Broadly speaking, LDA is a generative probabilistic model to analyze the latent topics from a set of documents or the data corpus. Using topic modeling, the document is

presented as a collection of latent topics and each topic is a collection of representative words. The LDA algorithm can be used to identify the latent topics from a set of documents by counting the word co-occurrence within the document. It then should conclude the number of distinctive topics (*K*) based on a coherence measure, which is defined as how well these topics "hang together" to represent the extracted latent topics [42]. After the most representative model has been trained through the iterative findings of the optimum setting of several parameters (discussed in the methods section), the LDA result will extract the most representative words in each topic and the distribution of those topics within the document. Eventually, we can interpret these distinct groups of words to understand the properties of topics (RQ1). According to this LDA result, and by carefully reading the content of representative documents, the term for each topic can be defined.

Mathematically, there are two matrices as the input and output of the LDA algorithm. The entry of a matrix row represents the distribution of word co-occurrence, as illustrated in Figure 1. The input matrix corresponds to the documents row (*D*) and the words column (*N*) across the entire dataset (dimension $D \times N$, *D* is the number of documents and *N* is the count of words), termed as "document–word matrix". Each entry of a document–word matrix represents the count of words co-occurred in each document. This input matrix will be modeled by the LDA algorithm to create two output matrices. They are a document–topic matrix ($\theta_D$) and a topic–word matrix ($\beta_K$) (Figure 1) that distribute the previous former document–word matrix using throughout a set of topics ($T_{1:K}$). The document–topic matrix ($\theta_D$) corresponds to the document rows (*D*) and the topic columns (*K*) (size $D \times K$, *D* is the number of documents and *K* is the number of topics). The entries of a $\theta_D$ matrix represent the co-occurrence of each topic within a single document. The topic–word matrix ($\beta_K$) corresponds to the topic rows (*K*) and the word columns (*N*) (size $K \times N$, *K* is the number of topics and *N* is the number of words). The entries of a $\beta_K$ matrix demonstrate the count of representative words in each latent topic. The interpretation of the LDA algorithm through this point of view is known as probabilistic matrix factorization, introduced by Hoffman et al. [43].

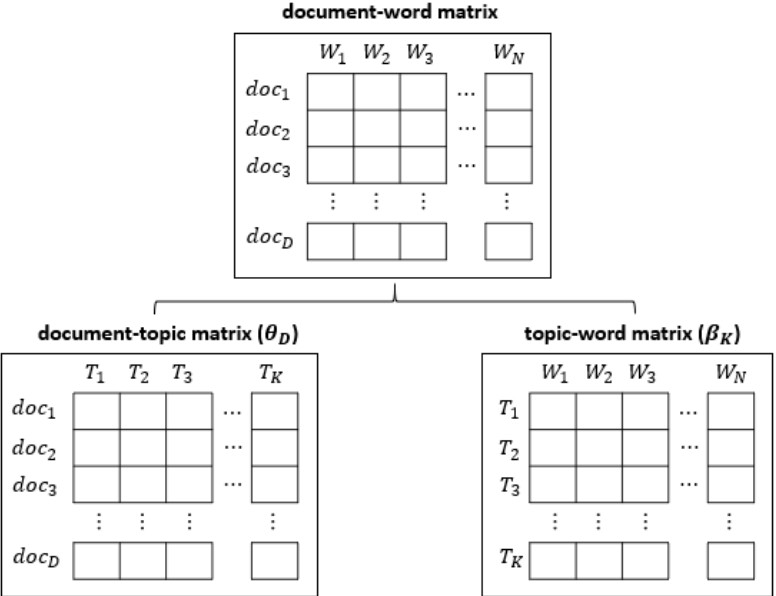

**Figure 1.** LDA interpretation through the concept of probabilistic matrix factorization (Adapted version from Odden, et al. [11]).

By the probabilistic matrix factorization, the LDA algorithm lies on three assumptions that must be taken into consideration by the user. The first assumption is that LDA does not consider the order of words in the analysis. Thus, it specifically disregards the nuance of language for text analysis. Indeed, it merely considers the number of words within the

document. Despite the existence of this major assumption, this is commonly assumed in topic modeling studies. As proposed by Grimmer & Stewart [19], the principle of text analysis is that "all quantitative models of language are wrong, but some are useful".

The second assumption of LDA is that all documents should contain a mixture of several topics rather than a single topic. Specifically, LDA believes in a mixed membership model of a topic, rather than a single model of topic contained in the document [44]. Fortunately, we argue that this second assumption should lead to the impactful merit of the LDA model in performing automated text analysis from the interdisciplinary nature including PER studies. We typically investigate specific research problems in PER. We often bring, share, and combine insights, theories, or methods from another related field. For instance, research-based physics instructions are evaluated through the administration of assessment tools validated in advance. In the interdisciplinary context, the PER community should consult several resources from curriculum and instruction studies and the field of educational measurement to support assessment validity.

The third assumption of LDA assumes that the representative words of a distinct topic will be more likely to be mentioned than another word within the data corpus. Then, this greater probability of a word in a topic means that that distinct word will tend to co-occur more frequently in each topic. This assumption is known as the distributional hypothesis of linguistics [18]. For instance, if the current topic of a document is "culinary recipes", the words belong to "food", "ingredient", "taste", or "cook" will be more frequently co-occurred rather than the less relevant words, i.e., "representation", "mechanics", "item", or even "conceptual understanding".

## 3. Method

Our study involved three common steps of LDA topic modeling, as demonstrated in Figure 2. In this section, we will explain the details of these stages consecutively.

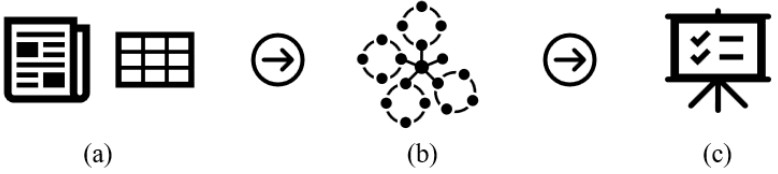

(a)    (b)    (c)

**Figure 2.** Common steps of the LDA study: (**a**) collecting and preprocessing the data; (**b**) modeling and evaluating the results; and (**c**) discussing the topical results to answer the research questions.

### 3.1. Collecting and Preprocessing the Data

In this step, we collected the PDFs by manually downloading the open access-based articles from five international conference proceedings between 2014 and 2021. Collectively, our dataset of Indonesian PER literature was sourced from 852 documents (~4 to 8 pages each). They were organized, peer-reviewed, and published by the Indonesian PER community. We involved the most five leading academic meetings within the Indonesian PER community including *International Conference on Research, Implementation, & Education of Mathematics and Science* (ICRIEMS) (*n* = 152) [45–54], *International Seminar on Science Education* (ISSE) (*n* = 220) [55–61], *International Conference on Mathematics & Science Education* (ICMSE) (*n* = 125) [62–69], *International Conference on Mathematics and Science Education* (ICMScE) (*n* = 291) [70–76], and *International Conference on Mathematics and Science Education* (ICoMSE) (*n* = 64) [77–80] to best capture the landscape of the Indonesian PER characteristics (RQ1) and their immediate development (RQ2). All those papers were published in the Scopus indexed proceedings (Journal of Physics: Conference Series by Institute of Physics (IOP) Publishing, Conference Proceedings by American Institute of Physics (AIP) Publishing, Advances in Social Science, Education and Humanities Research by Atlantis Press), and web-based repository of each conference hosted by the universities. Those conferences inevitably had multidisciplinary topics with other STEM education research.

Thus, we should ensure the downloaded file must be relevant to the PER aims only. In each conference, there was a clear section in which to choose the PER cluster.

We decided to analyze those conference proceedings since they are the oldest international conference organizers among the Indonesian Teacher Education Institutions (TEIs) and even within the Indonesian PER community. Furthermore, most of the authors were affiliated with several Indonesian TEIs and had various research experiences (graduate students to PER experts), and somehow attracted a few authors from neighboring Southeast Asian countries. The nature of "international" conferences inevitably had to involve non-Indonesian authors even if the conferences were organized by Indonesians. One can argue that these led to the misinterpretation that the currently selected papers failed to represent the Indonesian PER landscape. Nonetheless, this perception should be invalid if we remember that they are organized, peer-reviewed, and published by Indonesian PER scholars or even discussed and presented during a parallel session in the seminar. Moreover, the representation of authors affiliated as Indonesian was still the largest group from the data corpus. The contribution of authors from neighboring countries cannot be avoided since they could implicitly influence the development of the established Indonesian PER literature. Hence, there would be a likelihood that these overseas authors could inspire us and they are cited by the Indonesian PER scholars in their papers.

Furthermore, the authors of those publications came from outside of the organizing committees and from several regions of Indonesia hence it could represent a wider snapshot of Indonesian diversity. Additionally, those articles had also been peer-reviewed throughout using robust processes until the accepted decision was endorsed by the committee of publication. This criterion applied to our dataset should satisfy the eligibility standards for publications within the Indonesian PER community. We must admit that the selected proceeding papers analyzed in this paper could be arguable among other potential papers in Indonesian PER literature, i.e., other conferences or even academic journals. We see, however, the promising area of these other Indonesian PER literature that can be engaged in future thematic analysis studies.

After the articles had been gathered, we extracted the PDFs as a collection of words in each document using the "pdfminer" library within the python programming language. Then, we followed the common steps of data cleaning processes using the "nltk" library [81] which were admittedly time-consuming processes in the text analysis study [82]. First, we checked the downloaded files to ensure that they were in a good condition to be scraped as plain texts. Second, we removed the section headers ('Abstract', 'Keywords', 'Figure', 'Introduction', 'Table', 'Method', 'Conclusion'), authors' names, affiliations, references, and acknowledgment sections (if any) from the individual PDFs. Third, we deleted the numbers, symbols, punctuations, and stopwords based on the English vocabulary using the "nltk" library. Finally, the preprocessed texts were tokenized into a list of single words in each document as our document-word matrix (see Figure 1).

After that, we employed the "gensim" library [83] for lemmatizing and finding the bigrams. Lemmatization is the procedure to find the stem of some words in favor of the same meaning. For instance, "student" and "students" in the previous tokenized results should be lemmatized as "student". We then looked for the frequently mentioned pairs of words within the dataset, bigrams. For instance, "conceptual understanding", "problem solving" "scientific approach", "critical thinking", and so on (see more examples in Table 1). Bigrams should be combined by an underscore connecting the tokens. Finally, we had a "bag of words" containing 199,578 raw words and bigrams with 10,109 unique words. The tenth most frequent words in this current unfiltered data corpus are illustrated in Figure 3 below with their word frequency and fraction in each document (division between frequency and total of documents). The top five words that often co-occurred through our data corpus are "student", "learning", "physic", "skill", and "concept". These representative words demonstrate the scope of PER literature has been satisfied in our dataset. Nevertheless, these frequent words should be filtered to make for more efficient computing time and to make the extracted PER topics more distinct.

**Table 1.** Characteristics of the Indonesian PER topics based on their most representative words.

| Topic Number | Top 10 Representative Word | Weight | Topic Name |
|---|---|---|---|
| 1 | critical_thinking | 0.053 | 21st-century skill |
| | st_century | 0.025 | |
| | ability | 0.020 | |
| | creative_thinking | 0.016 | |
| | information | 0.014 | |
| | technology | 0.012 | |
| | data | 0.011 | |
| | communication | 0.011 | |
| | creativity | 0.010 | |
| | need | 0.008 | |
| 2 | test | 0.053 | Assessment |
| | assessment | 0.036 | |
| | instrument | 0.032 | |
| | item | 0.019 | |
| | level | 0.017 | |
| | question | 0.014 | |
| | ability | 0.013 | |
| | measure | 0.012 | |
| | development | 0.009 | |
| | analysis | 0.008 | |
| 3 | scienc | 0.034 | Interdisciplinary aspect of physics education |
| | eeducation | 0.019 | |
| | scientific_literacy | 0.015 | |
| | thinking_skill | 0.013 | |
| | thinking | 0.012 | |
| | ability | 0.012 | |
| | school | 0.012 | |
| | knowledge | 0.012 | |
| | scientific | 0.010 | |
| | level | 0.009 | |
| 4 | misconception | 0.031 | Conceptual understanding |
| | understanding | 0.030 | |
| | representation | 0.017 | |
| | conception | 0.010 | |
| | conceptual_understanding | 0.010 | |
| | scientific | 0.010 | |
| | level | 0.009 | |
| | phenomenon | 0.009 | |
| | difficulty | 0.009 | |
| | science | 0.008 | |
| 5 | model | 0.032 | Research based instruction |
| | activity | 0.021 | |
| | science_process | 0.018 | |
| | inquiry | 0.011 | |
| | achievement | 0.011 | |
| | class | 0.010 | |
| | science | 0.010 | |
| | learning_outcome | 0.010 | |
| | scientific | 0.009 | |
| | knowledge | 0.008 | |

**Table 1.** *Cont.*

| Topic Number | Top 10 Representative Word | Weight | Topic Name |
|---|---|---|---|
| 6 | problem | 0.035 | Problem solving |
| | problem_solving | 0.028 | |
| | ability | 0.023 | |
| | knowledge | 0.012 | |
| | solve_problem | 0.011 | |
| | improve | 0.010 | |
| | understanding | 0.010 | |
| | problemsolving_skill | 0.009 | |
| | approach | 0.009 | |
| | model | 0.009 | |
| 7 | medium | 0.037 | Educational technology |
| | development | 0.022 | |
| | material | 0.021 | |
| | technology | 0.017 | |
| | use | 0.016 | |
| | education | 0.010 | |
| | online | 0.009 | |
| | school | 0.008 | |
| | teaching_material | 0.008 | |
| | module | 0.008 | |
| 8 | experiment | 0.020 | Physics laboratory |
| | course | 0.013 | |
| | laboratory | 0.012 | |
| | motion | 0.010 | |
| | method | 0.010 | |
| | experimental | 0.009 | |
| | tool | 0.009 | |
| | practicum | 0.008 | |
| | understanding | 0.007 | |
| | activity | 0.007 | |

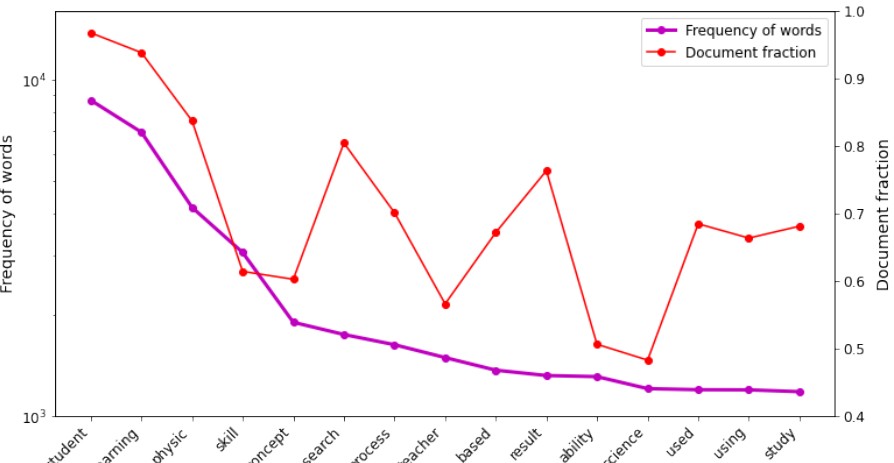

**Figure 3.** Distribution of the unfiltered word co-occurrence. The left axis represents the count of words and the right axis visualizes the document fraction within the unfiltered dataset.

Before we model the bag of words using the LDA algorithm, the next filtering processes for the most frequent and the rarest words should be followed. These words make our topical results difficult to identify. We want to discover unique terms to distinguish the research topics. Thus, the following step of data filtering was removing the most frequent words, and the rarest words that co-occurred within the bag of words. This removal action should be substantially noticed because the most-mentioned words might obscure the

character of the topic studied in the literature. Our extracted topics should be concerned with the most specific words rather than the most frequent words. Thereafter, removing the rarest co-occurring words would also make our dataset more efficient. The larger size of the data corpus with many noises (typos, names, locations, specific terms) would extend the running time of the LDA algorithm, hence the process will become inefficient. Several selections of the filtering parameters should be evaluated to achieve the optimum coherence value (described below). This process should be exhaustively repeated to ensure the most representative topics with the optimum coherence measure. A detailed description of the coherence measure will be explained in the next subsection.

In this paper, we elected to exclude the most frequent words whose frequency was greater than 55% within the dataset. Furthermore, we also excluded the rarest words whose frequency was less than eight times within the data corpus. They were selected based on several evaluation processes to obtain the most optimum coherence measure. Admittedly, this selection was also inspired by the previous practices of thematic analysis by Odden, et al [11]. Obviously, it eliminated a substantial number of unique words and bigrams, approximately 7724 words. Then, we had the cleaned data as many as 2385 total words and bigrams for the next LDA analysis. This was actually a huge number of removals, but they did not contribute towards distinguishing the specific description of a topic [82]. As explained above, this filtered dataset would make the modeling time of the LDA algorithm more efficient since it would mathematically reduce the dimension of the LDA matrices (see Figure 1). These filtering processes decreased the size of our dataset from 10,109 to 2385 unique words and bigrams (see Figure 4). These filtered versions of the dataset determined the final LDA model of the Indonesian PER topics which were evaluated by multiple iterative modeling processes based on the mixtures of the number of topics (*K*), hyperparameter *α*, and random initialization (seed number) to obtain the most coherent topics within the literature. Furthermore, these topics must be qualitatively evaluated by PER experts to strengthen the solid topical description based on their experiences.

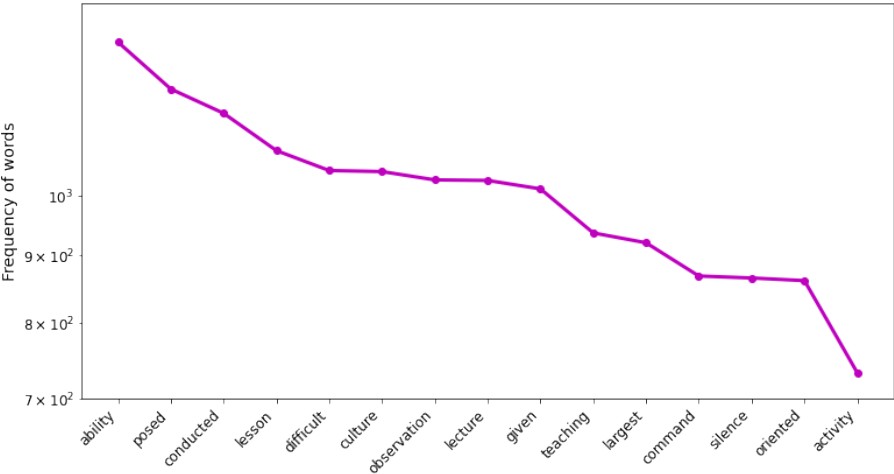

**Figure 4.** Distribution of words co-occurrence after the filtering process to the most frequent and the rarest words.

### 3.2. Modeling the Indonesian PER Topics through the LDA and Evaluating the Results

After the cleaned dataset had been served, we maintained it as a "pickle" file. Therefore, it could be imported directly without running the former code of data preprocessing and filtering processes. In this step, we conducted the iterative LDA modeling of the data corpus. The unsupervised nature of LDA requires us to manage several procedures of the evaluation process to find the final and the most representative LDA results. We must guarantee that their results make sense and do not deviate significantly based on the actual story of the research practice within the Indonesian PER field. In practice, users often implement one or multiple methods of evaluation to examine the LDA results [19,31,84]. Several

pieces of literature have described some possible methods of evaluation. Accordingly, this study considered two choices of evaluation methods from the literature i.e., coherence score and face validity. In this subsection, the iterative processes of tuning the final LDA model are described through these two evaluation processes.

### 3.2.1. Coherence of Descriptors in Identified Topics

Essentially, the coherence value is defined as an external evaluation metric of how mixed the descriptors (the most representative words) are in each topic. In other words, this measure quantifies whether these descriptors in each topic have supported each other to represent the topics. Basically, this is recommended by the distributional hypothesis of linguistics which believes that there must be some central words in a certain topic. The set of words in a single topic will occur differently in another topic [18,31]. Hence, this will measure how we can distinguish the extracted topics from the diverse set of words within the data corpus. Coherence values will be normalized between 0 and 1. The LDA results can be concluded as "more coherent" when it raises a higher value and is near unity [42]. The best value of coherence will determine the final set of filtering processes above and several hyperparameters that will be tuned in training the best LDA result.

Several hyperparameters that should be tuned during the iterative process of LDA modeling are the alpha ($\alpha$), random seed number, and the number of topics ($K$) [42]. Alpha is a hyperparameter that determines the relative "mixedness" of topics extracted by LDA. Moreover, the previous study has considered the potential issue during the training of LDA model, namely the random initialization seed [11]. It could cause a significantly different set of topics extracted from a single LDA model. Therefore, the LDA results are recommended to be interpreted from multiple random seed numbers. To find the most optimum model based on the coherence measure, we should train a high amount of LDA model in considering the mixtures of different numbers of topics ($K$), alpha ($\alpha$), and random seed number. In this study, we selected a mixture of eleven numbers of topics (4 to 14), five alpha values (1, 5, 7.5, 10, 12.5), and ten selected different seed numbers. The different seed numbers were inspired by the method of repeated measurement in the physics laboratory [85]. The calculation of coherence values is represented by the moving dots in Figure 5 around the average coherence value (red dot). From these combinations, we trained 550 LDA models represented by the spread of coherence values in Figure 5.

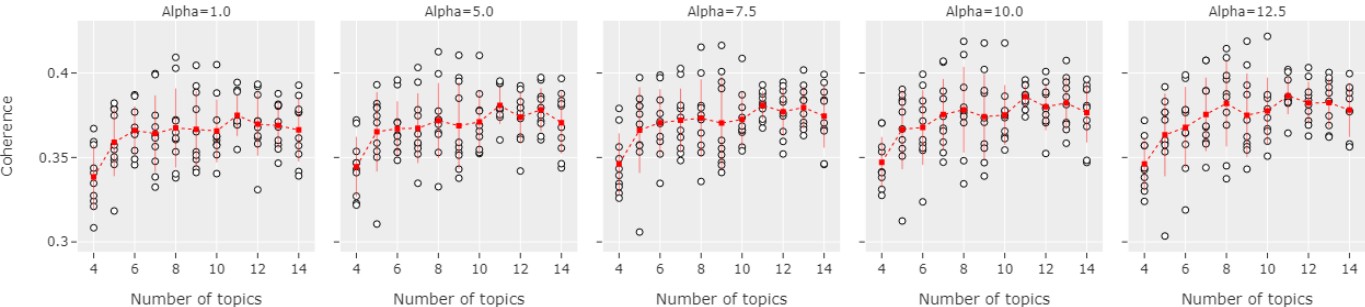

**Figure 5.** Coherence score ($\alpha$) within the mixture of number of topics ($K$), alpha ($\alpha$), and random seed number.

Using an elbow plot, Figure 5 is provided to summarize the behavior of our LDA model within these combinations. The spread of white dots in this figure are the varying coherence values within a single LDA model of a certain number of topics. Our obtained coherence values are between 0.31–0.42 as an acceptable measure for the results of the LDA model reported by the previous studies [11,12,20]. The red marker visualizes the average value from the variation of each $K$-value and their respective standard deviations. To determine the best selected parameters for the final LDA model, we employed the "elbow" method as suggested by the previous literature [31]. The best model would be diagnosed by the flat pattern from the elbow plot in Figure 5. We can see that coherence values are

greater with the increased number of topics and there is a leveling off pattern between six to 10 topics. This pattern can be an indication of diminishing returns. Based on these results, we choose the center of this range, *K* = 8, as our selection of the number of topics (*K*) for our final LDA model. This selection should be accompanied by the subsequent face validity from the PER experts to empower its representativeness within the literature.

3.2.2. Evaluating the Face Validity to the PER Experts

Face validity is a procedure to qualitatively evaluate the LDA results from the PER experts that are experienced with the established PER publications within the community. This will make sure the representativeness of our results based on their expertise and experience [86]. More technically, face validity requires experts who are familiar with the publication of the Indonesian PER field to judge how coherent the LDA results are based on their expertise, knowledge, and experience [31]. The second author of this paper is a professor in the Indonesian PER field with more than 20 years of research experience, particularly in the assessment and evaluation of physics problem solving and higher order thinking skills (HOTS). The third author of this paper is an associate professor of electrical engineering that has more than 20 years of research and teaching experience in programming language and artificial intelligence (AI) studies. These two authors confirmed the extracted topics that have been analyzed using the LDA algorithm. The second author presents to interpret the PER aspect and the third author contributes to guaranteeing our LDA algorithm in extracting the PER topics reported by this paper.

*3.3. Answering the Research Questions Based on the Final Trained LDA Model*

After the final LDA model has been trained to the most optimum coherence value, it will show the topical results derived from the data corpus. The aim of our study is to answer the two proposed research questions based on the most representative LDA model. This topic modeling results (see the next section) are then interpreted either to answer the proposed research question of the study or to re-evaluate the optimum model during the LDA training. The final model was trained from multiple phases of trial and evaluation toward different tuning of parameters described above. These processes should be exhaustively iterated in accordance with the most coherent results. After we discovered the coherence has been optimum, the final tuning of the LDA model would be selected.

In RQ1, the interpretation of the LDA model was explained in two ways. First, LDA results were understood by carefully examining the most representative words in each topic. In Table 1, we provide the top ten words of each topic. Our interpretation of these would be confirmed when these words have made sense based on the face validity. Accordingly, we can enumerate these results as eight research themes. Second, once the name of distinctive Indonesian PER topics had been determined, we then performed the subsequent strengthening interpretations to explore the most influential papers in each topic (see Table 2). In this table, we merely provide the five best representative papers of each topic to maintain the readability of this paper. In fact, we considered fifty representative papers in each topic to further study the characteristics of eight Indonesian PER topics. This analysis was necessary to obtain the next face validity to the extracted topics as well as to define the clear definition of the topic. Thereafter, the final terminology of each topic was decided according to these two steps of consideration. In RQ2, the evolution of each topic between 2014 and 2021 was measured by the "prevalence" parameter. In this study, the prevalence was defined as the percentage of each topic in each year within the collection of the annual documents [11]. A highly prevalent topic may be greatly studied in certain years but less focused on in other years. Eventually, it will illustrate the clear evolution of Indonesian PER studies for seven years that have been attempted. These results visualized what has been worked on by the Indonesian PER community and the potential room for future studies that could be addressed in the further journey.

## 4. Results

### 4.1. Characteristics of the Indonesian PER Topics between 2014–2021 (RQ1)

A final trained LDA model was employed to describe the characteristics of eight distinct Indonesian PER topics. They are reported in Table 1 with their representative set of words and in Table 2 with their representative set of papers in each topic as our baseline to interpret the LDA results and to understand how the Indonesian PER community has attempted the academic works. In this section, we will describe them in a consecutive way with supplemental interesting visualization in Figure 6 below.

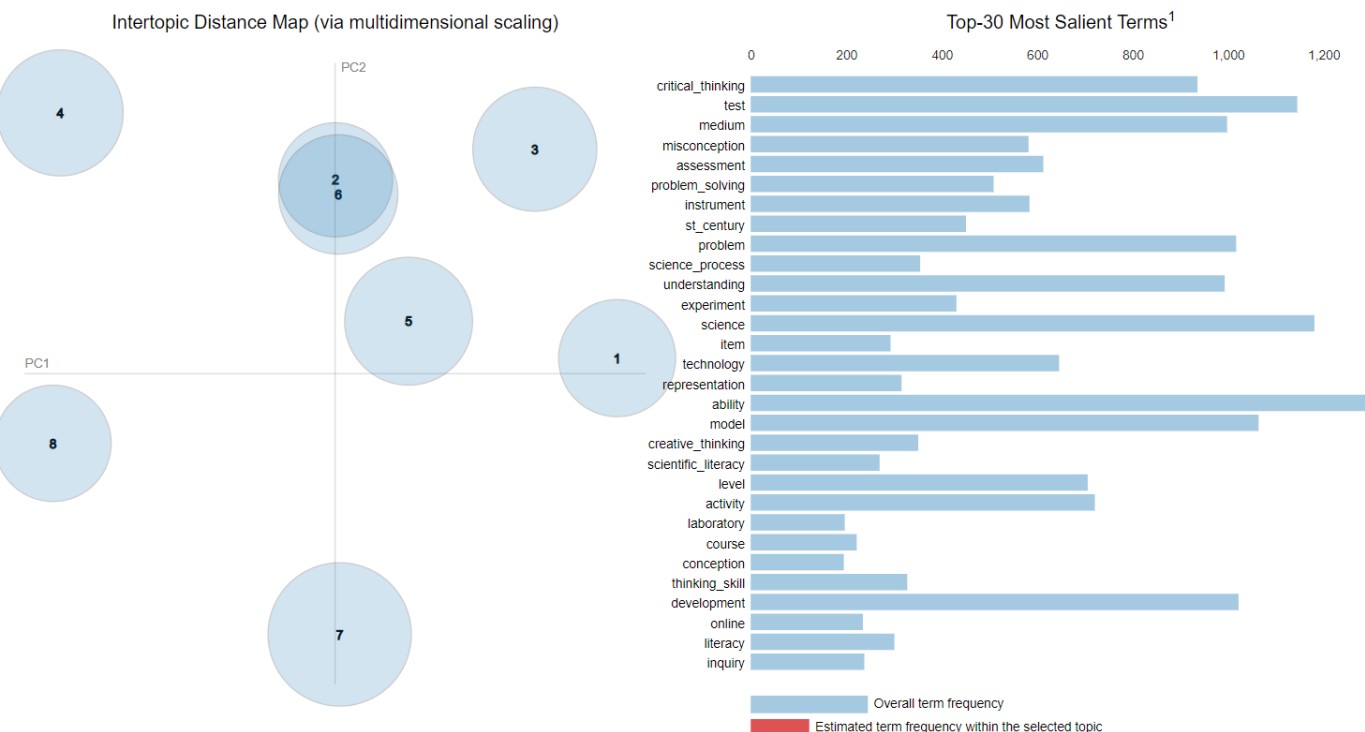

**Figure 6.** This figure is designed interactively thus if we select one of these thirty most salient words, we will obtain certain influential topics that are highly constituted by this word. This is the implementation of the distributional hypothesis of linguistics performed by the LDA algorithm. For instance, if we select "critical_thinking", this figure will make the zoomed bubble in the largest circle of topic 1 (21st century skill), smaller one of topic 3 (an interdisciplinary aspect of PER), and several tiny dots in other topics. It can illustrate that these zoomed topics (topic 1 and topic 3) have closely connected to each other and small dots at other topics have little connection to these topics. In this example of "critical_thinking", Indonesian PER researchers approached this skill as influential as 21st century skill and other interdisciplinary factors. ([1] Salience measure is calculated based on Chuang, et al. [87]).

As the first procedure of interpretation, we should initially notice the most representative words and weights of each topic number in Table 1. Essentially, the LDA results have no results about the research themes extracted from the literature. In practice, we situate Table 1 as being read from the left column to produce our interpretation of the topical name in the right column. It is implied that the right column of Table 1 is produced by the left part of the results. Our topic weights were probabilistic representations of each word in each topic which will become more relevant once the value is greater than a certain topic. Our findings report the spectrum of topic weights between 0.8% and 5.3%, which was also reported as acceptable measures by previous studies [11,20]. The order of the topic number is arranged based on the greater weight that represents how mixed the topic is within the literature.

Using the "pyLDAvis" library provided by the python programming language, the relationship among the Indonesian PER topics can be determined in Figure 6. In this study, we characterized eight distinct Indonesian PER topics studied between 2014 and 2021. The size of the displayed bubble in Figure 6 represents the most influential research theme within the Indonesian PER literature, namely "educational technology". The distance between the bubbles articulates the relative relationship of the topical results within a set of documents. We relatively found clear differences among the eight Indonesian PER topics produced in Figure 6. Even though "educational technology" has attracted the greatest focus of Indonesian PER scholars, it must be noted that the inter-topic distance map is constructed based on the multidimensional scaling of principal components (PC) emerging in the corpus. It is often assumed that it can be projected as a two-dimensional figure, as presented in Figure 6. Through this simplified visualization, we are assisted in illustrating the inter-topic relation that could be present among the emergent topics. This can be translated as the interdisciplinary nature of PER studies, as explained above.

Obviously, this will lead us to understand the disciplinary network that emerged within the PER community. Topic 2 (assessment) is closely correlated with topic 6 (problem solving). We suspect that this pattern is produced because the Indonesian PER scholars tended to develop and administer measurement tools to promote one form of students' performance, namely problem solving. Topic 2 (assessment) is also closely located with topic 5 (research-based instruction). It can be understood as the necessary evaluation metric after the implementation of several transformed physics learning within the PER community. Assessment must be required to measure the extent to which our physics learning reforms have effectively improved the students' learning process. To complement these aims, several students' performances from the national call of Indonesian curriculum are presented around these topics, including topic 1 (21st century skill) and topic 3 (an interdisciplinary aspect of physics education). In the next description, we will understand why this 21st century skill is connected to topic 6 (problem solving). This skill is one factor that should influence the critical and creative thinking of students as well as scientific literacy promoted by an interdisciplinary aspect of physics education. The advancement of technological development recently encourages students to contribute more to perform more sophisticated modern learning in 21st century society. These five topics can be clustered in quadrant I (positive x axes, and positive y axes) with their shorter relative distance from each other rather than the remaining topics, i.e., topic 4 (conceptual understanding), topic 7 (educational technology), and topic 8 (a physics laboratory). The separated relative distance from the quadrant I topics can be understood as the uniqueness of these topics within the analyzed literature.

The greater weight of the most representative words in Table 1 represents the more mixed the topics should be within the literature. Nevertheless, instead of Table 1, we recommended that one must interpret based on the most representative papers in each topic as further presented in Table 2. We admit that Table 1 can be troublesome since there are likely disconnected words of a topic, particularly in the case of small weights and making the interpretation trickier. Therefore, we supplement it by qualitatively crosschecking the content of the most representative papers on each topic in Table 2. This manner of literature reading is different from a traditional content analysis that was approached by the previous Indonesian author in [13–17]. Instead, we were aided by the topical results from Table 1, thus we merely explored the characteristics of each topic based on our clustered understanding in Table 1. In Table 2, we provide the prevalence, which is a quantitative measure of how mixed the paper is within a certain topic. For example, the 0.875 prevalence of Supahar's paper [88] in Table 2 articulates that it is composed of 87.5% of the assessment topic and the remaining values are lasid on the other mixture across all other topics. After the presentation of these tables, we detail the distinctive ways to differentiate the Indonesian PER topics that consider our results in Tables 1 and 2. This will justify the reason for which we interpret LDA results towards eight Indonesian PER topics.

**Table 2.** Representative articles, author, year, respective conference, and prevalence in each Indonesian PER topic.

| Topic | Article | Author | Year | Conference | Prevalence |
|---|---|---|---|---|---|
| 21st century skill | Profile of students' critical thinking ability in project-based learning integrated science technology engineering and mathematics | Eja, Ramalis, & Suwarma [89] | 2019 | ICMScE | 0.812 |
| | Gender differences in digital literacy among prospective physics teachers | Rizal, et al. [90] | 2020 | ICMScE | 0.799 |
| | Profile of senior high school in-service physics teachers' technological pedagogical and content knowledge (TPACK) | Masrifah, et al. [91] | 2018 | ICRIEMS | 0.776 |
| | Developing creative thinking skills of STKIP weetebula students through physics crossword puzzle learning media using eclipse crossword app | Anggraeni & Sole [92] | 2019 | ICMScE | 0.771 |
| | Evaluation of critical thinking skills of class x high school students on the material of Newton's laws | Febriana & Sinaga [93] | 2020 | ICMScE | 0.759 |
| Assessment | Applying content validity ratios (CVR) to the quantitative content validity of physics learning achievement tests | Supahar [88] | 2015 | ICRIEMS | 0.875 |
| | An eight-category partial credit model as very appropriate for four-tier diagnostic test scoring in physics learning | Istiyono, et al. [94] | 2021 | ISSE | 0.873 |
| | Developing of Bloomian HOTS Physics Test: Content and Construct Validation of The PhysTeBloHOTS | Istiyono, Dwandaru, Muthmainah [95] | 2019 | ICRIEMS | 0.866 |
| | Instrument test physics-based computer adaptive test to meet the Islam economic community literature review | Ermansah, et al. [96] | 2016 | ISSE | 0.861 |
| | Implementation of Item Response Theory at Final Exam Test in Physics Learning: Rasch Model Study | Asriadi & Hadi [97] | 2020 | ISSE | 0.858 |
| Interdisciplinary aspects of physics education | Mapping of professional, pedagogical, social, and personal competence of senior high school physics teachers in Yogyakarta special region | Jumadi, Prasetyo, & Wilujeng [98] | 2014 | ICRIEMS | 0.772 |
| | Analysis of Scientific Literacy Through PISA 2015 Framework | Arsyad, Sopandi, & Chandra [99] | 2016 | ICMScE | 0.766 |
| | Shifting attitude from receiving to characterization as an interdisciplinary learning toward ecological phenomena | Napitupulu, et al. [100] | 2017 | ISSE | 0.735 |
| | Promoting metacognition and students' care attitude towards the environment through learning physics with STEM | Rahzianta & Purnama [101] | 2016 | ISSE | 0.708 |
| | Analysis of senior high school students' higher order thinking skills in physics learning | Maulita, Sukarmin, & Marzuki [102] | 2018 | ICRIEMS | 0.690 |
| Conceptual understanding | Alternative conception of high school students related to the concepts in the simple electric circuit subject matter | Wardiyah, Suhandi, & Samsudin [103] | 2018 | ICMScE | 0.879 |
| | Identification of student misconception about static fluid | Setiawan, Saputra, & Rusdiana [104] | 2018 | ICMScE | 0.874 |
| | External representation to overcome misconception in physics | Handhika, et al. [105] | 2015 | ICMSE | 0.870 |
| | Teachers, pre-service teachers, and students understanding about the heat conduction | Anam, Widodo, & Sopandi [106] | 2018 | ICMScE | 0.869 |
| | Identify students' conception and level of representations using five-tier test on wave concepts | Wiyantara, Widodo, & Prima [107] | 2020 | ICMScE | 0.849 |

**Table 2.** *Cont.*

| Topic | Article | Author | Year | Conference | Prevalence |
|---|---|---|---|---|---|
| Research based instruction | The effectiveness of local culture-based physics model of teaching in developing physics competence and national character | Suastra [108] | 2015 | ICRIEMS | 0.846 |
| | Cooperative learning model design based on collaborative game-based learning approach as a soft scaffolding strategy: preliminary research | Nurulsari, Suyatna, Abdurrahman [109] | 2016 | ICMScE | 0.783 |
| | Effect of free inquiry models to learning achievement and character of student class XI | Kaleka [110] | 2018 | ICRIEMS | 0.773 |
| | Training students' science process skills through didactic design on work and energy | Ramayanti, Utari, & Saepuzaman [111] | 2017 | ICMScE | 0.769 |
| | The effects of cooperative learning model think pair share assisted by animation media on learning outcomes of physics in high school | Astra, Susanti, & Sakinah [112] | 2019 | ICMScE | 0.765 |
| Problem solving | The effect of e-learning based worksheet to improve problem solving ability of senior high school students | Septiyono, Prasetyo, & Ihwan [113] | 2020 | ISSE | 0.812 |
| | The analysis of students' problem-solving ability in the 5e learning cycle with formative e-assessment | Yuliana, et al. [114] | 2019 | ICoMSE | 0.797 |
| | The development of physics e-book based on contextual teaching and learning to increase student problem-solving skill | Fitriadi, Latumalukita, & Warsono [115] | 2021 | ISSE | 0.791 |
| | Improving students' problem-solving skills through quick on the draw model assisted by the optical learning book integrated the Pancasila | Himawan & Wilujeng [116] | 2019 | ISSE | 0.785 |
| | Profile of problem-solving ability of Islamic senior high school students on momentum and impuls | Sakti, Wilujeng, & Alfianti [117] | 2021 | ISSE | 0.766 |
| Educational technology | Developing whiteboard animation video through local wisdom on work and energy materials as physics learning solutions during the covid-19 pandemic | Anggraini, et al. [118] | 2020 | ISSE | 0.874 |
| | Android-based carrom game comics integrated with discovery learning for physics teaching | Rahayu, Kuswanto, & Pranowo [119] | 2020 | ICRIEMS | 0.864 |
| | Development of physics mobile learning media in optical instruments for senior high school student using android studio | Aji, et al. [120] | 2019 | ISSE | 0.843 |
| | Smartphone-based learning media on microscope topic for high school students | Nadhiroh, et al. [121] | 2020 | ISSE | 0.831 |
| | Android for the 21st century learning media and its impact on students | Adi, et al. [122] | 2016 | ISSE | 0.825 |
| Physics laboratory | Simple vertical upward motion experiment using smartphone based phyphox app for physics learning | Janah, Ishafit, & Dwandaru [123] | 2021 | ISSE | 0.865 |
| | The Atwood machine experiment assisted by smartphone acceleration sensor for enhancing classical mechanics experiments | Listiaji, Darmawan, & Dahnuss [124] | 2020 | ICMSE | 0.853 |
| | Development of sound wave experimentation tools influenced by wind velocity | Maisyaroh, et al. [125] | 2019 | ISSE | 0.840 |
| | Analysis of simple harmonic spring motion using tracker software | Mu'iz, et al. [126] | 2017 | ICMScE | 0.827 |
| | Real laboratory-based learning using video tracker on terminal velocity | Ristanto, Novita, & Saptaningrum [127] | 2016 | ISSE | 0.824 |

### 4.1.1. Topic 1: 21st Century Skills

This topic is the most mixed cluster based on the descending order of the weight measures of topical results. Promoting 21st century skills is discovered as the main con-

cern from papers published within the Indonesian PER community. Keywords including "critical_thinking", "creative_thinking", and "communication" are several components of students' performances in 21st century learning. Students are expected (refers to "need") to grasp the well-known four components of 21st century learning skills (4Cs) [128]. Additionally, the abundance of digital technology in the past few decades encourages our physics educators to approach their physics learning with digital platforms represented by the terms of "information", "data", and "technology". It is undoubtedly also connected with the focus of the seventh topic below (educational technology). The vast development of the digital age during this century motivates physics educators to be concerned in this area. Therefore, this topic could be stated as the most influential party and increasingly takes much attention within the Indonesian PER literature for the past few years.

The research questions studied under the 21st century skill topic are predominantly made up of several categories: technological developments for physics learning and laboratory reforms in promoting 21st century skill [92,129–140], small- to large-scale survey in evaluating physics learner performance on this skill [89,90,141,142], correlational study toward another form of students' performance [143–145], and designing measurement tools to probe this skill on physics learning and instruction [146–150]. One could consider that this vast amount of literature is closely connected with other topics discussed below. For instance, technological development in this topic overlaps with the seventh topic (educational technology), and the emergence of physics laboratories in this topic is closely connected with the eighth topic (physics laboratory), and obviously with the second topic (assessment). Nevertheless, we argue that the uniqueness of the current topic is underlined by the focused aims to address the modern idea of 21st century learning. It promotes 21st learner skills including creative thinking [92,137–139,143,144,146,150], critical thinking [89,131,134,142,143,145–148,150,151], collaborative problem solving [130], data literacy [132,133,135,136], and digital literacy [90,152]. Moreover, Indonesian PER scholars are also attracted to approaches beyond high school physics instruction. Several studies have attempted to support pedagogical competence for professional physics teachers [153] or even prospective physics teachers [90,154–157]. These efforts can be made to ensure the physics educator as a mastermind of the physics classroom has to collectively support the intention of 21st century physics learning. Thus, they are expected to engage with this vision in physics learning responsively.

### 4.1.2. Topic 2: Assessment

This topic focuses on developing, validating, and disseminating measurement tools that are needed in performing assessments throughout the physics learning process and evaluating research-based instructions within the PER community. It is composed of several representative words for which we designed and developed measurement tools including "test", "instrument", "item", "question", and "measure". These tools are disseminated to define the quantitative measure of "ability" within physics learning or students' performance in physics classrooms. Moreover, several modern measurement theories including item response theory and Rasch modeling are mainly discussed by the Indonesian PER members within this topic. The emergent "level" keyword can be related to the other topics below, particularly with the third and fourth topics of our topical results. It could articulate several assessment concerns to factors that were mainly highlighted on students' performance within the Indonesian PER community.

In this second topic, several measurement tools have been developed and disseminated within the Indonesian PER community. They are comprised of performance tests and diagnostic tests. Performance tests are designed to measure diverse forms of students' performance on physics learning, including cognitive test [97], higher order thinking skills (HOTS) [95,158,159], critical thinking skill [160], representation [161–166], data literacy [167], digital literacy [168], science process skills [169,170], problem solving skills [171,172], inductive thinking [173], visual literacy [174], communication skills [175], analytical thinking skills [176], and scientific literacy [177]. Moreover, several diagnos-

tic tests are also established by the Indonesian PER authors to detect potential students' misconceptions [178–182], lack of representation ability [183–185], lack of higher order thinking skills (HOTS) [186], lack of critical thinking skills [160], lack of problem-solving skills [184], lack of data literacy [187], as well as the lack of understanding throughout astronomy class [188]. On the other hand, one can argue that this topic seems to be similar to the other extracted topics currently discussed. For instance, in this topic, we discover that several research-based assessments (RBAs) are addressed to measure 21st century skills. They are critical thinking, data literacy, digital literacy, and problem solving. Additionally, the same set of physics learning skills emerged as discussed further in the third topic (an interdisciplinary aspect of PER) and the fifth topic (problem solving). We argue that this second topic can be distinguished from other topics in its focus on the dissemination of the robust methodology to design, examine, and evaluate the developed measurement tools for physics education. Several validity studies have been introduced including content validity [88], factor analysis [175], Rasch model [97,188–191], and engaging modern measurement theory of dichotomous and polytomous response model [94,192] from item response theory (IRT). Additionally, our RBAs are designed through several mediums including computer aided tests [160,193], computerized adaptive tests [194], two- to six-tiered tests [160,180–182,188], and other forms of the test let [195].

### 4.1.3. Topic 3: Interdisciplinary Aspect of Physics Education

The topic of 21st century skill guides the Indonesian PER scholars to a focus on the interdisciplinary aspect of physics learning. Physics can be studied as an integral part of science, engineering, technology, and mathematics (STEM) education. Physics should be taught to understand complex understanding about contextual phenomena. The phase of the 2013 Indonesian curriculum oriented the physics teachers to engage the philosophy of "scientific approach" in their learning [196]. Due to our dataset being drawn from 2014 to 2021 literature, it is reasonable when this topic can be situated to address the implementation of this ongoing curriculum. We enumerate this topic as an interdisciplinary aspect since the nature of physics education during this timeframe should involve an "integrated" understanding of science. Physics is closely connected with other STEM subjects such as mathematics, biology, and chemistry. The terms "science" and "education" can emerge within this topic due to most of the Indonesian PER studies believing that their physics learning should be adjusted to solve contextual phenomena using physical knowledge supplementing with another scientific knowledge. For instance, Napitupulu, et al. [100] engage ecological phenomena assumed as crucial factors to which physics education should address. Moreover, physics education can be transformed to harness moral values about the environmental aspects of the ecological issue. Using a metacognitive framework, Rahzianta and Pratama [101] support the previous idea of Napitupulu, et al. [100] that physics education can foster the value of awareness toward environmental attitude. Through physics instruction, students were also expected to be critically aware of the challenge about the integrated issue of science education.

Essentially, the 21st century skill topic above inevitably correlates to this movement in preparing physics students to face the future complex challenge of their modern real world. Students are expected to acquire several skills that they learn through physics learning in terms of scientific literacy (refers to keywords "scientific_literacy", "knowledge", "scientific"), higher order thinking skills (HOTS) (refers to "thinking_skill" and "higher_order") [197–199], and another form of "thinking" processes [200–206]. Research movements on scientific literacy in this topic can be driven by the international announcements of *Programme for International Student Assessment* (PISA) assessment for Indonesian secondary students [207–217]. PER members are one of discipline-based education research (DBER) on STEM education (refers to keywords "science", "education", "school") that is responsible for this duty call in improving students' performance on PISA results. In addition to the focus of this topic, the keyword of "thinking_skill" is particularly relevant to "higher_order" in the eleventh rank of representative words in this topic, nevertheless, it

could not be shown in Table 1. Higher order thinking skills (HOTS) are also considered as part of other students' performance that are associated with other factors including scientific literacy and the first topic (21st century learning skills) [218,219]. Furthermore, unique findings from the Indonesian PER literature are discovered in promoting character values through physics education [220–223].

4.1.4. Topic 4: Conceptual Understanding

This topic is relevant to the previous results in Docktor and Mestre's [9] synthesis results of international PER literature for several decades. The earliest movement of PER literature underlined conceptual understanding as fundamental for physics learning. Docktor and Mestre [9] place this topic as the first theme of their thematic results. Our findings can be different from the results reported by Docktor and Mestre [9] since our conceptual understanding is discovered as the fourth topic. As previously described, the Indonesian PER community is encouraged mostly to the first topic (21st century skills) due to the national call for a scientific approach curriculum (2013 curriculum). Nevertheless, conceptual understanding could not be ignored from the Indonesian PER development. Indeed, we must admit that this topic is still imperative for physics learning among the other students' thinking skills and problem solving skills formerly mentioned. The name of conceptual understanding could be concluded in this topic because there are several representative keywords in Table 1 including "misconception", "understanding", "conception", and obviously the bigram of "conceptual_understanding". Using the LDA topic modeling, Yun [12] also recognized this current topic as an "introductory physics" theme in their results toward data corpus from *The American Journal of Physics* (AJP) and *Physical Review Physics Education Research* (PRPER). The keyword "conceptual" in Yun's results emerged in the first theme extracted from the AJP dataset.

Furthermore, "representation" of students' understanding is considered as a specific form of conceptual physics understanding [9,11]. Odden et al. [11] even discovered "representation" as their first topical results extracted from the same methodology of LDA algorithm. The term "difficulty" in conceptual understanding is also studied in our result. Likewise, other interdisciplinary aspects of physics understanding, such as "scientific", "phenomenon", and "science" emerge because of our movement to the third topic above. As discussed earlier, conceptual understanding of topics obviously influences other topics within the data corpus. The term "level" interestingly occurred in this topic as mentioned in the third topic (assessment).

One of the research questions explored in this topic is identifying conceptual knowledge about physics performed by Indonesian students [106,107,224–227] or physics teachers [106,228–230]. They investigated conceptual physics understanding on mechanics [107,227,231,232], electricity [224,228,230], magnetism [226], fluid [104,229], work and energy [225,233,234], thermodynamics [106], and modern physics [235]. Within the context of the Indonesian PER literature, we propagate conceptual understanding in another form of multiple representations [225,236,237], including external representation [105], mental model [238], drawing ability on free-body diagram [239,240], and mathematical representation [232].

Furthermore, diverse difficulties also have been discovered within the literature [227,234]. Various terminologies have emerged from Indonesian PER literatures to define the students' lack of understanding about conceptual physics, namely alternative conception [103,241], misconception [104,231,233,235,236,242–250], and misunderstanding [251]. To address this limitation on students' conceptual understanding, the Indonesian PER scholars have designed and examined vast learning reforms or interventions, i.e., conceptual construction-reconstruction oriented instruction (CCROI) [252], remedial programs [253], authentic learning [254], cognitive conflict instruction (CCI) [243], electronic conceptual development conceptual change text (E-CDCCText) [255], conceptual change-oriented text (CCO-Text) [235,248], and conceptual change laboratory (CC-Lab) [256]. Their purpose is to address students' misconceptions thus students can be supported to follow the conceptual progression [252,255], learning progression [253], or conceptual change [235,254,256].

Eventually, studying conceptual understanding through correlational inquiry has also been worthwhile to conduct [241].

4.1.5. Topic 5: Research Based Instruction

In improving the students' performance (refers to "achievement", "knowledge", "learning_outcome") on physics learning, several learning transformations and curricular developments (refer to "model", "activity", "class") have been attempted by the Indonesian PER members in this topic. As briefly discussed above, due to the national call of the 2013 curriculum, Indonesian physics education during this timeframe was encouraged to approach science process skills as five cycles of learning paces in physics learning. The paces include observing, asking questions, experimenting, explaining (or reasoning), and presenting (or reporting) abbreviated as "5M" in the Indonesian language [257]. They can be translated as inquiry-based learning in practice. Our fifth topic makes sense if the Indonesian PER literature mentioned keywords including "science", "science_process", "inquiry", and "scientific" in this topic. The term "activity" in the LDA results also implied that the "scientific" approach recommended by the 2013 Indonesian curriculum was engaged in students' activities within physics learning. Admittedly, they are also closely connected with the interdisciplinary topic on the third of our LDA results above.

We consider that this topic is one of the most diverse groups within our LDA results. Nonetheless, most of them are essentially designed based on the philosophical lens of constructivist learning. Indonesian physics education has a long history of adopting the student-centered learning approach since the establishment of the 1968 Indonesian curriculum [258]. We have probed several students' performance on the physics learning approaches above. Research-based instruction is generally designed and implemented to promote them through constructivist learning. On the other hand, we discover distinct aspects derived from the Indonesian PER literature that cover studies to approach the indigenous, cultural, or local context of Indonesian physics learning. Several learning reforms were inspired by culturally relevant aspects of Indonesian diversity, as reported by Suastra [108]. This learning tradition makes different colors emerge in Indonesian physics education besides the five scientific cycles-oriented learning approaches in the implementation of the 2013 national curriculum. They are reported by diverse papers, particularly in addressing inquiry-based learning [259–262], project-based learning [263–265], and problem-based learning [266–268]. Moreover, Indonesian PER scholars are motivated to adapt physics learning through the lens of a cooperative framework (social learning theory), i.e., collaborative game-based learning [109], think pair share (TPS) [112], time token [269], and social learning cycle [270].

4.1.6. Topic 6: Problem Solving

Relevant to the fourth topic above, this topic is also precisely reported by Docktor and Mestre's [9] synthesis analysis. They discuss this topic as the second position of their thematic result. Currently, our LDA model discovers several terms in this topic related to problem solving definition, including "problem", "problem_solving", "solve_problem", and "problemsolvingskill". In supporting students' success in physics learning, apart from the conceptual understanding discussed above, problem solving skills (several termed as ability) is also a fundamental factor to be a successful physics learner. Content knowledge of physics is primarily discovered through critical problem-solving steps to explore and understand how our physical circumstance works. Moreover, several terms including "improve", "approach", and "model" represent that the Indonesian PER scholars propagate it as a learning strategy to endorse this imperative topic as recently discussed in the fifth topic. They cover particularly the implementation of a problem-based learning model. Eventually, physics education could contribute to improving problem solving skills that inevitably correlate with 21st century skills for students' future.

As described in other studies focused on students' learning, this topic mainly commenced with the profiling of students' performance in solving physics problems [117,271–275]. These

reports can be cited as a basis for Indonesian PER scholars to develop physics learning reforms [116,276–280], curricular developments [113,281–284], and computer-aided instruction [285,286] to improve the Indonesian students' performance in physics problem solving. Contextual issues within Indonesian society were on several occasions engaged with by the Indonesian PER authors, including cultural context [287] and disaster mitigation awareness [285,288–290]. The immediate movement of this contextual learning is grounded on physics as an interplay within STEM education. Therefore, physics educators have great expectations that students can learn complex thing from physics and make concrete efforts within their social communities.

### 4.1.7. Topic 7: Educational Technology

Admittedly, the first topic of our LDA results has been tremendously influenced by the emergence of this seventh topic within the Indonesian PER literature. The keyword "medium" in this topic is lemmatized from "media" during the preprocessing step of the LDA modeling. Physics instruction is motivated to follow the disruptive effect of the digital age in the 21st century era. The existence of digital technology makes our learning transform in response to these circumstances. We discover that this topic is frequently mentioned in several papers with regard to developing learning material (refers to keywords "material", "teaching material", "module") through technology-enhanced learning (refers to "technology", "online") implemented in physics classrooms. Broadly speaking, technology can be flourished from the manifestation of our understanding of science. Digital technologies, i.e., computers and mobile devices, have tremendously encouraged Indonesian PER scholars to be involved in physics learning and instruction. Complex applications within education makes this topic definitely diverse and broad. The demand for 21st century learning, the national call of the 2013 curriculum, and the rapid development of the digital age have been impactful for Indonesian PER scholars in the development of a vast number of technical assistances within physics learning, including audio-visual media [118,291–298], web-based applications [299,300], android applications [119–122,301–308], augmented reality [309], and distance learning [310–312]. The cultural context of Indonesian society is presented through the delivery of educational technology [306,313–319]. The former interdisciplinary aspect of physics education and the demands of 21st century learning drives an intention during the design and implementation of educational technology on physics [304,320,321].

### 4.1.8. Topic 8: Physics Laboratory

In Table 1, we discover several keywords bringing us to the definition of this topic as our learning scheme within the physics curriculum. Experimental physics is considered as one vital path through which physics knowledge might be taught to all physics people. We name this topic as a physics laboratory since "experimental" physics learning typically occurs in the laboratory setting. This topic focuses on how physics learning or "course" can be delivered through real [125,322–326] or virtual "laboratory" [327–331] in conducting the physics experiment (refers to "activity" and "practicum"). Several papers also have developed their own physical measurement "tool" and data acquisition using microcontrollers, trackers, or smartphones [124,332–337] that could be employed to enhance students' experience within physics laboratories. Eventually, through this channel, PER studies also consider addressing their learning transformation to improve "understanding" of physics [338,339]. The appearance of the keyword "motion" in this topic represents that a physics topic is mostly addressed on Newtonian mechanics as also reported by Yun's results based on *The American Journal of Physics* (AJP) journal [12].

### 4.2. Development of the Indonesian PER Topics between 2014 and 2021 (RQ2)

In the second research question, we investigate the development of the extracted Indonesian PER topics between 2014 and 2021 through the measure of topic prevalence. We adopt the definition of prevalence that has been approached by a previous study by

Odden, et al. [11]. Prevalence of a particular topic is defined as the sum of documents that are categorized on that topic within the amount of literature published in a certain year. This measure is represented as a percentage that could be aggregated both cumulatively (Figure 7) and averaged (Figure 8) by year. For instance, a 10% prevalence of topic 1 in a certain year has a two-fold meaning. First, it represents the average prevalence of topic 1 for that year as many as 10%. Then, the cumulative prevalence of topic 1 for that year is its multiplication with *n*, in which *n* is the number of documents published in that year. If the annual cumulative prevalence of all topics is summed up, then it would correspond to the total of documents published in that year.

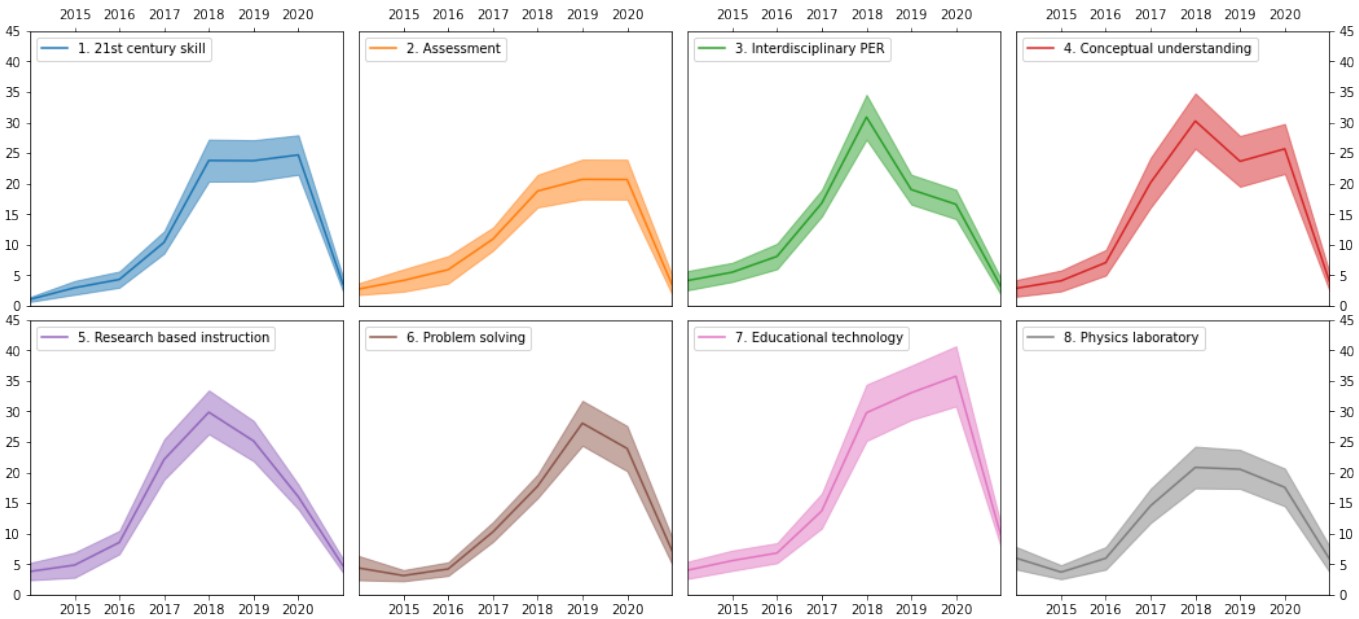

**Figure 7.** Cumulative prevalence of Indonesian PER topics development between 2014 and 2021.

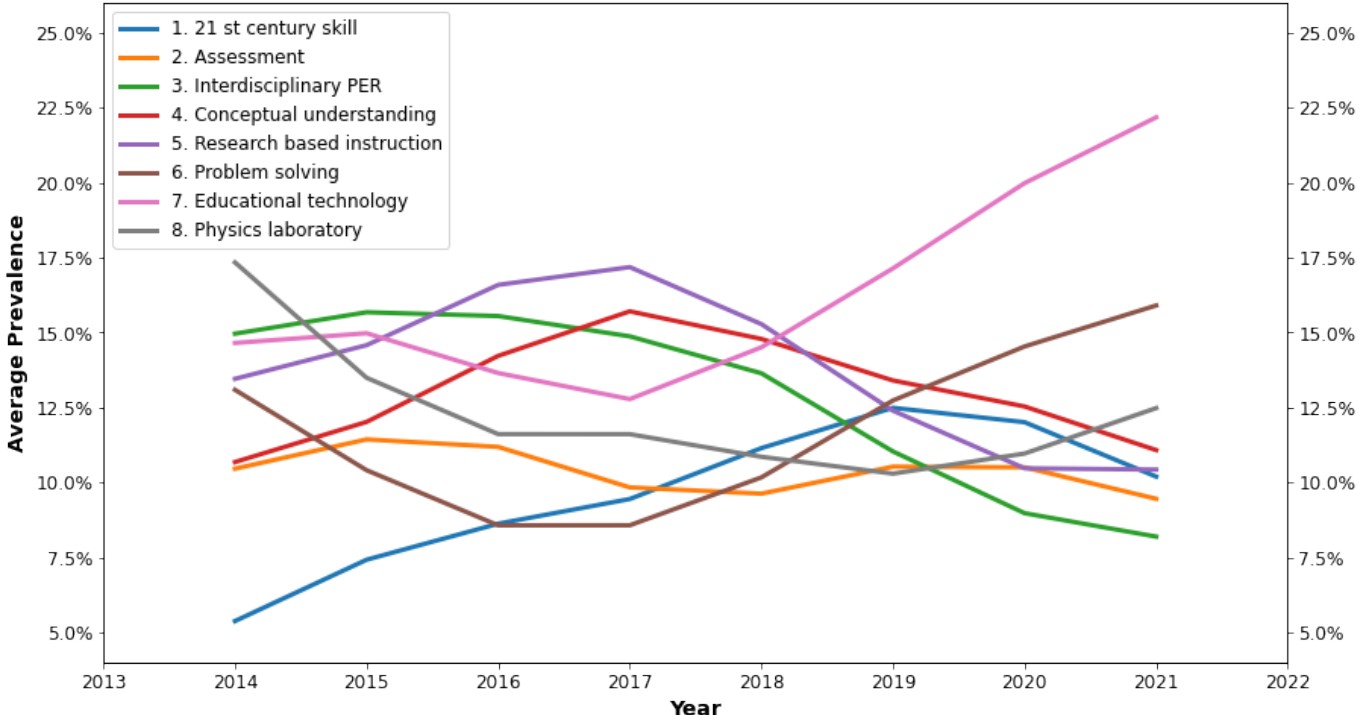

**Figure 8.** Average prevalence of the Indonesian PER topics development between 2014 and 2021.

The cumulative prevalence of eight Indonesian PER topics between 2014 and 2021 is illustrated in Figure 7. The cumulative prevalence of a topic in the $y$ axis is provided as the number of "effective" papers disseminated in that year. For example, 25 cumulative prevalence of 21st century skill topics in 2018 (see Figure 7) means that there are equivalent to 25 "effective" articles discussing about 21st century skill topics in that year. This term "effective" is inspired by the previous research [11] because, keeping in mind, LDA results underlie the assumption of the mixed membership of topics. An individual article should be categorized into several topics (in varying weights) rather than a single topic.

We provide shaded areas in Figure 7 to describe the topical distribution within the annual topic development. The width of the shaded area in Figure 7 is the standard deviation ($\sigma$). We use as many as $3\sigma$ from the mean value represented by the solid line in the figure. We calculated this standard deviation using the jackknife resampling technique [340]. For certain topics and years, this procedure yielded a new sample of 100 cumulative prevalence values. Using this newly generated sample, the standard deviation is calculated to describe the distribution of a topic prevalence in each year. The jackknife resampling method described above produces the shaded areas that could be represented as the topical variation for a certain year. A shaded area of zero for one year would be produced if there is no difference among the cumulative prevalence of several topics during a single year. On the other hand, if there are several papers that are focused heavily on a certain topic, the shaded area (topical spread) would be larger.

Figure 7 illustrates that our whole topics have demonstrated relatively similar rise and fall between 2014 and 2021. There is a spike in 2018 and 2019 followed by a decrease in the subsequent year for all topics. We suspect that the apparent decrease can be driven by several publications in the year 2021 that are still progressing. Broadly speaking, the disruptive transition during the 2020 pandemic year has tremendously influenced the attendance of potential PER researchers from several parts of Indonesian institutions [341]. Moreover, our dataset for 2021 conference is merely sourced from the ISSE conference and the rest of the conferences are still progressing through publication processes. Figure 7 describes the lowest cumulative topic prevalence that occurred in the early year of 2014. The finding is not surprising because there were only two conferences that have been organized by UNY (through ICRIEMS) and UNNES (through ICMSE) in that year. A measure of cumulative topic prevalence is particularly dependent on the number of documents written for a particular year. There is stable cumulative prevalence particularly on 21st century skill and assessment topics even though the assessment topic has a lower prevalence. Educational technology has had the highest increased prevalence for the past few years. There are similar spikes described by the interdisciplinary aspect of physics education and conceptual understanding topics in 2018. However, for the following year after this, the interdisciplinary PER topic has a more substantial decrease than a conceptual understanding topic. Problem solving topics have the latest spike in 2019. Unfortunately, the physics laboratory seemed to be a minority within the Indonesian PER community due to the smallest topic prevalence among other Indonesian PER topics.

As described above, the cumulative measure of topic prevalence is merely dependent on the number of "effective" documents published in that year. Regarding the relative number of papers published in a certain year, an average measure should be defined. It could be fairly utilized to compare different topics from year to year. In the calculation of an average measure, we can employ the data-smoothing technique which dampens the effect of sample dependence in the year-to-year variation. In this study, we choose the three-year rolling windows that will average the prevalence values for each year with those of the former and the subsequent year. Figure 8 depicts our plot of average Indonesian PER topics prevalence over time.

Based on the average prevalence visualization in Figure 8, there is the relative stability of rising and falling for all the topics between 2014 and 2018. The most interesting topics within the literature are interchanged over years. In early of 2014, the physics laboratory topic emerged to dominate the movements, however, this topic follows a decreasing

pattern through several subsequent years after that. In the next year, the interdisciplinary aspect of physics education has attracted our Indonesian PER scholars for their attention within the community. We suspect that the increasing pattern of the third topic must be motivated by the governmental policy of the 2013 curriculum. Moreover, there is a continuous pattern that research-based instruction topics lead the waves between 2016 and 2018. Nevertheless, this topic has substantially decreased in the subsequent years and the position is overtaken by educational technology topics after 2018 and problem-solving topics after 2019. We then notice that the assessment topics remain stable over time on average. The assessment of physics learning is inevitably a multidisciplinary field within educational science. Measurements of students' performance and validation studies using various methods, either from classical or modern theory, are still needed for the development of discipline-based educational research (DBER) including the Indonesian PER community. Furthermore, it then indicates that this PER topic has been studied through collective development to support the promotion of 21st century skill and other students' performance including interdisciplinary aspects of PER, conceptual understanding, and problem solving. In the early years, it is interesting that 21st century skills even had the lowest attention in 2014. Although we cannot conclude where this trend comes from. Looking at the representative papers on this topic (see Table 2), we argue that the lowest prevalence of 21st century skill in the early year of 2014 corresponded to the limited digital technology that has been approachable during this year. Eventually, this topic will continue to develop until 2019. It is likely to become greater in following the associated trends of increased educational technology until 2021.

## 5. Discussion

In this paper, we have demonstrated that the LDA algorithm from NLP, a subfield of ML studies, offers a potential tool to analyze the plethora of publications within the Indonesian PER community. For the answer to RQ1, we have extracted eight Indonesian PER topics using the LDA algorithm toward the selection of five publications on physics education research conferences organized, peer reviewed, and published by Indonesian PER members between 2014 and 2021 [1–5]. They are composed of (1) 21st century skills, (2) assessment, (3) interdisciplinary aspects of physics education, (4) conceptual understanding, (5) research-based instruction, (6) problem solving, (7) educational technology, and (8) physics laboratory. The description with the representative references to distinguish each of these emergent topics has been provided through Tables 1 and 2 above with a description of representative papers to emphasize our understanding of the topics.

Furthermore, Figures 7 and 8 above have been provided to enrich our insights about the development of Indonesian PER studies since the beginning of 2014 to date. For the answer to RQ2, the development of the Indonesian PER topics has dominated interchangeably over this timeframe. Nevertheless, we admit that several topics recommend that their development appear fair and stable between 2014 and 2021. In the early years of our analysis period, Indonesian PER members put their attention more towards studying how physics learning should be immersed through a physics laboratory. Thereafter, we discovered that it was overtaken by research-based instruction in transforming physics learning into several reforms to approach various forms of student performance that are constructed based on the interdisciplinary understanding of physics education. In more recent years, the Indonesian PER field has been encouraged by the demand for digital technology-enhanced learning that attracted Indonesian PER scholars to develop teaching aids for physics instruction using various technological approaches. This was also relevant to the movement of problem solving topics during the time to promote the increasing trends on 21st century learning since 2014.

We can discuss these current findings by comparing them to those previous works that have been published before our paper [9,11,12]. Table 3 summarizes PER themes that have been reported by Docktor and Mestre's review [9], Odden et al. study [11], and Yun's thematic analysis [12]. Some topics from our findings are found to be in common in

these previous works, but some topics can be distinct. Using more traditional large-scale synthesis analysis, Docktor and Mestre have extracted PER topics into six primary topical areas of physics education research. Using the same method as the current study, Odden et al. have extracted PER topics into ten research themes based on 1302 individual papers published in the physics education research conference (PERC). Additionally, eight PER themes were also extracted by Yun [12] based on the data corpus from AJP and PRPER journals using a similar methodology to our paper (LDA algorithm). From these three references, we will discuss how our Indonesian PER findings show immediate points of overlap or several unique patterns different from the previous works.

**Table 3.** Previous works about characteristic and development of PER topics within the community.

| Docktor and Mestre [9] | Odden et al. [11] | Yun [12] | |
| --- | --- | --- | --- |
| | | AJP | PRPER |
| 1. Conceptual understanding<br>2. Problem solving<br>3. Curriculum and instruction<br>4. Assessment<br>5. Cognitive psychology<br>6. Attitudes and beliefs about teaching and learning | 1. Representation<br>2. Problem solving<br>3. Labs<br>4. Quantitative assessment of concept<br>5. K-12<br>6. Difficulties with quantum mechanics<br>7. Community, identity<br>8. Qualitative methodology and constructivist theory building<br>9. Research based instruction<br>10. Quantitative survey of demographic gap | 1. Introductory physics<br>2. Teaching models<br>3. Force and motion<br>4. School program<br>5. Problem solving<br>6. Pedagogical content knowledge<br>7. Students' learning strategy<br>8. Experiment | 1. Assessment<br>2. Gender<br>3. Student's concept<br>4. Teacher education<br>5. Students' reasoning process<br>6. School programs<br>7. Introductory physics<br>8. Problem solving |

One can technically compare our topical findings in Table 1 to the previous works in Table 3. There are several topics or themes that are overlapped and are more distinctive. We have three similar findings precisely to Docktor and Mestre's [9] review on conceptual understanding, problem solving, and assessment topics. There are three topics overlapped with Odden, et al.'s [11] thematic analysis including problem solving, physics laboratory (labs), and research-based instruction. Yun's [12] results from AJP analysis exactly match our topical results on teaching models (research-based instruction), problem solving, and experiments (a physics laboratory). From PRPER findings of Yun's results, we demonstrate three relevant research themes including assessment, students' concept (conceptual understanding), and problem-solving topic.

These topical results are followed by three unique Indonesian PER topics that are missing from three previous studies. They are 21st century skills, interdisciplinary aspects of physics education, and educational technology. We argue that these immediate differences correspond to the different contexts according to the authors' point of view. If we review synthesis results of Docktor and Mestre [9], those three different topics might be categorized in the context of assessment or curriculum and instruction. Educational technology that has been developed by Indonesian PER members is assumed as a learning transformation within the PER community summarized in Docktor and Mestre's "curriculum and instruction" theme. Moreover, 21st century skills and interdisciplinary aspects of physics education are highly motivated by the Indonesian educational context, 2013 curriculum, and PISA results as explained above. They engage other forms of students' performance considered in the assessment topic of Docktor and Mestre's results. Moreover, this unique pattern derived from Indonesian PER literature can be understood as educational development within a certain country that should be determined through several social contexts and governmental policies [208,258,342,343].

Furthermore, based on Odden, et al. [11] topical findings, our unique findings can be illuminated by the topic of K-12 based education. In this scope of the theme, high school physics contributes to developing our third topic, the interdisciplinary aspect of physics learning. The scientific approach-based Indonesian 2013 curriculum inevitably directed physics educators to orient interdisciplinary high school (K-12) physics learning. The Indonesian PER community is tremendously conducted by the preparation of high

school physics teachers on the national need for sustainable physics teaching and learning. Development of PER dissemination can be indirectly seen to respond to this national call. Several educational technologies have been developed by PER scholars to make the delivery of physics learning more engaging to all students from all backgrounds.

Moreover, we can discover other similar topics with different theoretical lenses from Yun's thematic analysis [12]. From her results, we highlight topics on force and motion, pedagogical content knowledge (PCK), students' reasoning process, and introductory physics. The latter is even reported by Yun both from AJP and PRPER journals. We found that force and motion is also the most interesting topic within the Indonesian PER community. In Table 1, we discover the keyword "motion" as the representative word to define the eighth topic, physics laboratory. Likewise, we discuss the relevant research on conceptual understanding and problem-solving topics addressing the concept of force and motion. We also believe that PCK and introductory physics can be related to each other to implement the transformation of physics learning. They are intended to deliver more effective physics learning for students. Therefore, we argue that these topics can have the same meaning as our fifth topic of the LDA results, research-based instruction.

For the open room of future projects, we argue that the Indonesian PER scholars should pay more attention to investigating physics education research more qualitatively. We argue that Indonesian PER topics should address research focused on qualitative aspects of physics teaching and learning as addressed by Docktor and Mestre's results as their fifth and sixth PER theme, Odden, et al.'s findings as their seventh and eighth PER theme, and Yun's inventions as their fourth topic from AJP results and their second and sixth theme from PRPER results. Compared to the Odden et al. thematic results, there are qualitative topics dealing with community and identity as well as qualitative methodology and constructivist theory building that are still missing within the Indonesian PER literature. Yun's topical results about gender and school program support these findings to grasp demographic factors within physics learning, including gender bias on physics assessment, students from underrepresented minorities or first generation, as well as supporting the vision of diversity in physics [344]. This methodological approach is also relevant to Docktor and Mestre's result to investigate cognitive psychology and attitudes and beliefs about physics education. Those trends still lack research within Indonesian PER literature and there is possible room for future study on this topic.

It is evident from our paper that the LDA algorithm has demonstrated several advantages in undertaking thematic analysis towards 852 Indonesian PER proceeding papers over time. We can describe its strength as two-fold explanations. First, the automation of the LDA algorithm inevitably has technically helped us to make classification of eight Indonesian PER topics without extra effort to manually scrutinize the data corpus. We also utilize almost the whole section of the body of the research paper. Thus, our current study can suggest that LDA considers the more comprehensive nature of thematic analysis rather than using the keywords from research titles as reported by Faisal [14] or selecting small parts of documents [13,15–17]. Second, the distributional hypothesis of topics and the mixed membership of topics have been satisfied through the LDA algorithm. These advantages have explained the existence of multidisciplinary aspects of physics education research. Categorization of a single topic in each document as reported by Faisal, et al. [14] and Bancong, et al. [13] fails to represent that each topic should be interchangeably in each document. Nevertheless, in nature, our paper is dedicated to the aim of exploration and attempts to deliver a promising tool to conduct a more efficient methodology of thematic analysis which successfully helps us to add dimensions of analysis and visualization. Traditional methods of thematic analysis must be worthwhile and cannot be replaced by the current methodology. Indeed, the LDA algorithm complements them to extract a more comprehensive understanding from thematic analysis.

On the other hand, we cannot forget the potential weaknesses after the implementation of the LDA model performed in this study. As discussed by previous work [11], there are admittedly several limitations of the LDA algorithm in the analysis of research literature.

First, LDA clearly neglects the sequence of words within sentences as clearly assumed in our theoretical review above. Our LDA results above are calculated based on the count of words occurring in the data corpus. Thereafter, the qualitative method of thematic analysis obviously can be more beneficial to address this issue. To address this first obstacle, evaluation methods through face validity with experts in specific domains (PER) should be attempted. Second, the instability of topical results is evident during the training of the most representative LDA model. This is driven by the random initialization of the computation of the LDA model. In order to address this second limitation, multiple LDA models should be trained across the mixture of several hyperparameters including a number of topics ($K$), alpha ($\alpha$), and several filtering parameters to the most frequent and the rarest words. In this study, we trained a high number of LDA models within eleven numbers of topics ($K$), five different alphas ($\alpha$), and we iterated ten selected different integers of our seed number. This produced 550 LDA models and then we chose the most optimum model based on the coherence measure using the elbow plot provided in Figure 5. Third, we discovered that LDA can be more sensitive to literature that has grown over a long period. Several specific topics that are not frequently mentioned within the data corpus cannot be detected in the results. Obviously, they are likely to be excluded based on our rule of filtering actions.

As a final mark, one can realize that our findings must be dedicated primarily to the Indonesian PER community. Since, to the best of our knowledge, similar research has never been attempted within the Indonesian PER community using the LDA to break down the growing size of Indonesian PER literature. Research institutions can adopt our topical findings to establish a solid definition for the research group of PER works. Subsequently, we hope it could encourage novice PER scholars to easily recognize the characteristic of the Indonesian PER and guide them to contribute to specific group within the community. Furthermore, our paper should recommend several topics that have been published and future directions that should be approached in the next research project within the community, particularly in the aspect of the qualitative methodology of physics education research. Through our LDA results, the Indonesian PER community can understand what valuable steps have been attempted and where the future Indonesian PER community must go. The LDA methodology demonstrated in this paper can inspire the wider Indonesian PER members to utilize this current method of thematic analysis. Admittedly, we cannot ignore that the results of this analysis may be interpreted as having a different meaning regarding other authors that accidentally did not publish their works at those conferences. The determination of five conferences that have been analyzed through our analysis might be an arguable position that has been selected by the authors. Ultimately, other PER researchers could look forward to using the LDA method for future explorations of the larger Indonesian PER literature in the next efforts.

## 6. Conclusions

In summary, Indonesian physics education research (PER) literature has been thematically analyzed using the LDA algorithm. Eight topics were attempted by our PER members including 21st century skills, assessment, interdisciplinary aspects of physics education, conceptual understanding, research-based instruction, problem solving, educational technology, and physics laboratory. In the early initiation of Indonesian PER conferences in 2014, our members placed more attention on approaching learning through physics laboratories. This brought us to the movement of the community in responding to the demands of 21st century learning experiences within physics lessons. Our educators then were encouraged to harness several educational technologies to promote several aspects of students' performance in physics and interdisciplinary aspects of physics education, including scientific literacy and higher order thinking skills (HOTS) based on the demand of 21st century learning. We can declare that the LDA algorithm has been demonstrated as a powerful computational tool to extract insights derived from Indonesian PER literature. The automation technology embedded in this algorithm made the literature review methodology through thematic analysis robust in terms of its findings for the merit of the

research community. Furthermore, this paper could be the basis to understand the extent to which Indonesian PER scholars have made efforts to develop their community to date. Our results may recommend future work that should be conducted within the community, particularly about the qualitative aspect of physics learning and instruction, that is little known according to the results reported in this study.

**Author Contributions:** Conceptualization, P.H.S. and W.H.; methodology, P.H.S.; software, P.H.S. and W.H.; validation, E.I. and H.; formal analysis, P.H.S.; data curation, P.H.S.; writing—original draft preparation, P.H.S.; writing—review and editing, P.H.S.; visualization, W.H.; supervision, E.I. and H. All authors have read and agreed to the published version of the manuscript.

**Funding:** This research received no external funding.

**Institutional Review Board Statement:** Not applicable.

**Informed Consent Statement:** Not applicable.

**Data Availability Statement:** The cleaned data of this project (pickle format), the python code of this thematic analysis and metadata of the articles (excel format) can be downloaded at: https://github.com/santosoph/Indonesian-PER-thematic-analysis (accessed on 19 July 2022).

**Acknowledgments:** This article is an ongoing part of a doctoral study on the Graduate School of Educational Research and Evaluation (PEP) which the first author (P.H.S.) is currently pursuing at Universitas Negeri Yogyakarta (UNY), Indonesia. We would like to express the highest gratitude to The Ministry of Education, Culture, Research, and Technology (KEMENDIKBUDRISTEK), The Center for Education Financial Services (PUSLAPDIK), and The Indonesia Endowment Funds for Education (LPDP) of The Republic of Indonesia for providing the Indonesia Educational Scholarships (BPI) so that the first author (P.H.S.) is able to pursue this academic degree.

**Conflicts of Interest:** The authors declare no conflict of interest.

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
