# Peer review of "Thematic Analysis of Indonesian Physics Education Research Literature Using Machine Learning"

_data_

Round 1
Reviewer 1 Report
The article uses LDA to discover the main topics in Physics Education Research (PER) done by Indonesian authors to study the prevalence and evolution of them. The Article has a correct methodology regarding the machine learning aspects, and has an interesting result, as it is in concordance with previous results. The Article has some areas of opportunity that can be addressed to enhance further the paper.
You mention that the novelty lies on using LDA, in this part of the introduction: “Our novelty of the study clearly could be stated that we have recommended LDA algorithm because it has potential ability to break down great number of Indonesian PER literatures” However, the article is neither a recommendation, nor the first use of LDA for finding topics in PER. For it to be a recommendation, you would need to compare against other methods, such as Latent Semantic Analysis and then discuss the pros and contras of each one. Which is not the main topic of the article. And the use of LDA for PER is used in Odden et al., which you cite. I think the merit of the paper lies on the use of the technique in a more diverse array of journals, and the verification and comparison with previous results (Docktor and Mestre, and Odden et al.).
It is ambiguous the Indonesian part of the research. As your collection of data section mentions that the selected conferences have attracted people from neighboring countries, and all the open access articles were obtained, it is not clear how that represents the Indonesian PER landscape. Were all the authors in the used papers exclusively from Indonesians universities? If now, how they were filtered? It is important to mention this to understand the applicability of your sample.
The article reaches makes some assumptions that probably are best avoided. For example: “Therefore, those publications might be able to represent topics studied on Indonesian PER field”, “This might surpass the eligible standards for publications within Indonesian PER community”, or “this unique pattern derived from Indonesian PER literatures could be 600 understood as educational development within certain country might be determined 601 through several social contexts and governmental policy”. Since these assumptions are outside of the scope of the study, and nothing is cited that supports them, they are better avoided.
Section 3.3.2 mentions that there are other methods for checking the validity but they have limitations, so they weren’t used. This paragraph is not useful, because it does not discuss the other methods, their relative merits, or what could be the negative effects of using them. Thus, it does not further the discussion. Please state which methods could be used for validation and for each one explain why it is not applicable.
Author Response
Point 1: You mention that the novelty lies on using LDA, in this part of the introduction: “Our novelty of the study clearly could be stated that we have recommended LDA algorithm because it has potential ability to break down great number of Indonesian PER literatures” However, the article is neither a recommendation, nor the first use of LDA for finding topics in PER. For it to be a recommendation, you would need to compare against other methods, such as Latent Semantic Analysis and then discuss the pros and contras of each one. Which is not the main topic of the article. And the use of LDA for PER is used in Odden et al., which you cite. I think the merit of the paper lies on the use of the technique in a more diverse array of journals, and the verification and comparison with previous results (Docktor and Mestre, and Odden et al.).
Response 1: Thank you for your valuable comment about the novelty of our paper. We have actually described it in the Introduction section about the justification on which our novelty of the study has been addressed. However, as stated by Alajami (2020), “it is difficult to recognize what the original scientific production is without specifying the position of originality in the atmosphere in which it is rooted as well as the ecosystems surrounding this original intellectual production or that.” Therefore, the novelty of our paper should be considered as a contextual issue surrounding how Indonesian PER has been established. In this response, the context of Indonesian PER ecosystem should be central to the careful consideration towards the novelty of our research.
We could understand how our research should be novel in three ways. First, the lack of representative Indonesian PER scholars is still present in the previous international works. In line with Alajami’s argument above, we have oriented the readers about the context that “Those great works admittedly have guided the PER community in several parts of the world. Nevertheless, the representative Indonesian scholars in these international publications were still limited to capture the Indonesian PER context. For that reason, it might be relatively less appropriate to understand the development of Indonesian PER topics if we merely consider those resources without eliciting the context of Indonesian PER scholars.” (line 77-81). Therefore, our extended intention to understand the development of Indonesian PER recommends that Indonesian contexts should be engaged even through the similar methodology performed by the previous results. We believe that seeing this issued context should be considered as the novelty of our research paper to enrich the merit of previous references. To address this first issue, in this study, we initiated to analyze 852 proceeding papers published within Indonesian PER community that have not been included in the previous references. Even it can be discovered that we are the first Indonesian PER scholars to contribute to this area.
Second, there is lack of understanding about the LDA method to conduct topic modeling within Indonesian PER community. As we mention in line 82, “Additionally, to the best of our knowledge, Indonesian PER researchers have not yet performed this similar work to analyze their literature.” In the context of international PER studies, we must admit that LDA was not the first used to extract our understanding about the evolution of PER community. Nonetheless, to date, there is no Indonesian PER researcher that has approached this methodology to extract the understanding of our literature. Most of the Indonesian PER researchers still used the traditional methodology to analyze their literature as reported by Ref 12-15. To promote LDA dissemination within Indonesian PER community, we have performed it in this study that is potentially to be reported to extract our understanding about Indonesian PER development. In addition to the first claim that we are the first to analyze these amount of data above, we believe that our LDA method performed in the paper should also be the first dissemination of LDA method in the context of the Indonesian PER researchers. Obviously, it will tremendously contribute to the development of our PER community. We hope our paper will trigger other Indonesian PER scholars to analyze the wider Indonesian PER literatures in the future.
Third, we have to admit that the conventional method of content analysis that has been implemented by other related fields with Indonesian PER scholars presents several limitations. We have to admit that empirical study about analyzing the published literature within Indonesian PER community is still missing (once again see line 82). The reasons have been explained in line 83-84 of our paper. Consequently, we approach several studies from other STEM disciplines in Ref 12-15 to give some examples of our “ecological sphere” surrounding the development of our community. Content analysis or traditional analysis that we refer to in this paragraph obviously show several limitations in its methodology. In line 87-107, the limitations of this approach have been discussed in detail. We compare it with the nature of research topics that should be attempted through LDA modeling that has been briefly introduced in line 108-118. Facing these limitations, we initiate to perform LDA modeling as our solution to the previous drawbacks that occurred on traditional method of thematic analysis. Eventually, these three reasons or justifications should underline how our research paper has been sufficiently novel for the context of Indonesian PER community.
To address this potential issue, we have made several improvements in our Introduction section, particularly in line 128-168. Moreover, we have made major improvement in the Results section to provide the more detailed context to the broader reader about Indonesian PER results in line 576-821.
Point 2: It is ambiguous the Indonesian part of the research. As your collection of data section mentions that the selected conferences have attracted people from neighboring countries, and all the open access articles were obtained, it is not clear how that represents the Indonesian PER landscape. Were all the authors in the used papers exclusively from Indonesians universities? If now, how they were filtered? It is important to mention this to understand the applicability of your sample.
Response 2: In accordance with your fourth comment, the Indonesian PER literatures (as covered in our title of paper) should be defined as research articles that have been organized, peer-reviewed, and published by Indonesian PER ecosystems. It can be implied since the Abstract section in line 9 in which the term “disseminated” is clearly mentioned. Dissemination of studies must be processed by the Indonesian PER community to be published and worthy to be read. We highlighted it again in line 19-20, 272-274, 301-303. Even though the name of conferences were framed in terms of “Mathematics and Science Education”, we have carefully chosen articles published in the PER section only in this study. We have described it in line 282-293. Therefore, the context studied outside Indonesian PER themes should have been avoided.
Indonesian PER literatures can be represented through five conference proceedings that have been chosen by our study. In line 44-51 of the introduction section, we have described how these oldest five conferences have been organized by four Indonesian Teacher Education Institutions (TEIs) since 2014. Admittedly, the nature of an “international” conference is to recognize non-Indonesian authors who are able to contribute throughout the conferences. Accordingly, one can argue that these can be misinterpreted and that the selected papers do not represent the Indonesian PER community. Nonetheless, these perceptions should be invalid if we could remember that they are organized, peer-reviewed, and published by Indonesian PER scholars or even discussed and presented during the parallel session in the seminar. Moreover, the representation of authors acknowledged as Indonesian were still most of the data corpus (~ 98%). The contribution of authors from neighboring countries could not be avoided since implicitly they must influence how our discourses shape our understanding within Indonesian PER literatures. There is a possibility these articles would be discussed and cited by Indonesian PER scholars in the future study.
To address this potential issue of ambiguity, we have made a new paragraph about our definition about Indonesian PER literatures, particularly in line 298-318.
Point 3: The article reaches makes some assumptions that probably are best avoided. For example: “Therefore, those publications might be able to represent topics studied on Indonesian PER field”, “This might surpass the eligible standards for publications within Indonesian PER community”, or “this unique pattern derived from Indonesian PER literatures could be understood as educational development within certain country might be determined through several social contexts and governmental policy”. Since these assumptions are outside of the scope of the study, and nothing is cited that supports them, they are better avoided.
Response 3: We have made some improvement to address this issue in line 319 and 325. Then, we argue that in line 943, we have addressed the different results of our research from the previous results. The reason can be explained as the contextual factors including social contexts and governmental policies ruled by certain countries.
Point 4: Section 3.3.2 mentions that there are other methods for checking the validity but they have limitations, so they weren’t used. This paragraph is not useful, because it does not discuss the other methods, their relative merits, or what could be the negative effects of using them. Thus, it does not further the discussion. Please state which methods could be used for validation and for each one explain why it is not applicable.
Response 4: We have excluded it as recommended by the reviewer. See line 405 for the description of evaluation method employed in the study.
References :
Alajami, A. (2020). Beyond originality in scientific research: Considering relations among originality, novelty, and ecological thinking. Thinking Skills and Creativity, 38. https://doi.org/10.1016/j.tsc.2020.100723
Faisal, Gi, G. M., & Martin, S. N. (2020). Analysis of government-funded research in Indonesia from 2014-2018: Implications for research trends in science education. Jurnal Pendidikan IPA Indonesia. https://doi.org/10.15294/jpii.v9i2.23174
Reviewer 2 Report
Here's what I took away from this paper: The authors used a topic modeling technique from NLP, Latent Dirichlet Allocation, to thematically analyze the recent trends in Indonesian physics education and science education research literatures. This analysis showed that this literature has seen some small waves of research interest over the last 9 years, most notably an uptick in research on educational technologies since about 2018. They compare the topics extracted by LDA from this body of research to those extracted by both a previous LDA-based study and a large scale traditional literature review, finding that several of the topics included in the Indonesian PER literature are not present in these other literatures and several of the most prevalent topics in the other two reviews were missing from the Indonesian literature base. The authors recommend that Indonesian physics education researchers investigate these topics in future research projects. I think the idea behind this project is valuable, and much of the analysis is strong. The topics derived by the researchers seem well-interpreted and explained, and their trends seem to make sense. I also appreciate their comparison between these topics and those in the wider PER literature, and their recommendations that researchers take inspiration from those topics not currently represented in Indonesian PER. However, there are several issues with the analysis and presentation that need to be addressed. The first relates to attribution. From reading this paper it is clear that the authors are using the analysis approach established by Odden et al. (2020), which is cited in ref 11: Odden, T.O.B.; Marin, A.; Caballero, M.D. Thematic Analysis of 18 Years of Physics Education Research 670 Conference Proceedings Using Natural Language Processing. Physical Review Physics Education Research 2020, 671 doi:doi.org/10.1103/PhysRevPhysEducRes.16.010142. It also seems clear that the authors are using the code base from that project as the starting point for their analysis, given how similar their figures are to those in that paper. If this is the case, it should be explicitly acknowledged in the article. Furthermore, the argumentative structure of the article, from the introduction to the specific sub-sections to the flow of the text, is strongly reminiscent of that article. In fact, at several points I had such a strong sense of deja vu reading this article that I had to go back to the article by Odden et al. and check to see if the text was, in fact, different. I do not believe that this quite reaches the level of outright plagiarism, given the fact that this analysis is using an established research methodology to analyze a different literature based to answer different research questions. However, I suggest the editorial team do some cross comparisons between the two papers and evaluate whether they feel the authors should keep the current argumentative structure (with additional attribution, given the degree to which this analysis seems to have built off the Odden analysis) or whether they should rewrite it to be more distinctive. Next, there are some issues that are technical in nature:- The authors seem to have conflated two different factors in LDA analysis, alpha and coherence. These are not, in fact, the same. Alpha is a hyperparameter that controls the relative "mixedness" (or, conversely, distinctiveness) of topics learned by LDA. Low alpha values will produce topics that are very distinct, while higher alpha will produce topics that are more similar. Coherence is an external evaluation metric that evaluates how well the topics fit with the data. More about coherence can be found in M. Röder, A. Both, and A. Hinneburg, Exploring the space of topic coherence measures, in WSDM 2015Proceedings of the Eighth ACM International Conference on Web Search and Data Mining (Association for Computing Machinery, New York, 2015), p. 399, https:// doi.org/10.1145/2684822.2685324. Because these two metrics are not the same, the authors both need to correct their explanation of them in the paper and (potentially) redo their analysis anywhere where they have conflated the two. Given the range of reported coherence scores I suspect that it is primarily a presentation issue, but that needs to be fixed
- In the authors' evaluation of coherence to find topic number, they only seem to have evaluated a single model per K-value. Coherence scores can vary quite widely based on the random seed initialization of the LDA model. I suggest the authors create additional models at each K-value (around 10 is often sufficient) and evaluate their coherence scores as well to see if there is significant variation
- It was unclear to me whether the authors are actually analyzing Indonesian PER or whether they are analyzing a larger literature base that just happens to include some Indonesian researchers. Given their research questions I would have expected the first, but the literature they use seems to come from a much wider variety of conference proceedings including papers both outside of Physics Education Research and authors outside of Indonesia. Given their title, abstract, and research questions, I suggest the authors elaborate on how they ensured that the literature base focused on Indonesian Physics Education Research or else modify their study framing to include this wider set of disciplines and nationalities.
- It was also unclear to me how the authors chose the random initialization seed for their final LDA model. This was a significant focus in the Odden et al. article, since different random seed values can produce significantly different sets of topics. It is good practice for LDA analyses to clarify the procedure used to choose this parameter. If they did not use any specific procedure, that is a flaw in the methodology
- Line 170: "Each entry in the document–word 170 matrix represents the probability of word mentioned in each document." These entries represent the count of the words, not the relative probability
- Line 314: " The descriptor of topics is “more 314 coherent” when alpha is more higher and near to unity [74]. " This is incorrect, as alpha and coherence score are different measures.
- Line 527: "Three rolling windows averaged the 527 prevalence values for each year with those of the former and subsequent year. " The authors likely meant "three-year rolling windows" since this is what was done in the Odden et al. analysis
Author Response
Point 1: However, there are several issues with the analysis and presentation that need to be addressed. The first relates to attribution. From reading this paper it is clear that the authors are using the analysis approach established by Odden et al. (2020), which is cited in ref 11: Odden, T.O.B.; Marin, A.; Caballero, M.D. Thematic Analysis of 18 Years of Physics Education Research Conference Proceedings Using Natural Language Processing. Physical Review Physics Education Research 2020, doi:doi.org/10.1103/PhysRevPhysEducRes.16.010142. It also seems clear that the authors are using the code base from that project as the starting point for their analysis, given how similar their figures are to those in that paper. If this is the case, it should be explicitly acknowledged in the article. Furthermore, the argumentative structure of the article, from the introduction to the specific sub-sections to the flow of the text, is strongly reminiscent of that article. In fact, at several points I had such a strong sense of deja vu reading this article that I had to go back to the article by Odden et al. and check to see if the text was, in fact, different. I do not believe that this quite reaches the level of outright plagiarism, given the fact that this analysis is using an established research methodology to analyze a different literature based to answer different research questions. However, I suggest the editorial team do some cross comparisons between the two papers and evaluate whether they feel the authors should keep the current argumentative structure (with additional attribution, given the degree to which this analysis seems to have built off the Odden analysis) or whether they should rewrite it to be more distinctive.
Response 1: Thanks so much for your valuable comment in arguing the originality of our research. First of all, we must admit that our paper is partially inspired from the Odden, et al (2020), particularly python code that we implement towards Indonesian PER literatures. In fact, we have personally requested permission from Odden via email in Jan 2021 before we intended to conduct the study. Please see the screenshots below that demonstrate our communication delivered to Odden. Clearly, he has provided permission to us in using their code as long as we make appropriate attribution as citation to the original work of Odden, et al (2020). We can see that we have mentioned it at Ref [11].
Additionally, we think that the same image style as presented by our manuscript can not be problematic as long as we provide appropriate citation to Odden, et al (2020). One can think that our case similarly occurred when we use the same commercial software in analyzing different data. Obviously, a similar image style would be produced. Nevertheless, it could not be judged as the same results precisely since the data sourced from different contexts. Broadly speaking, we should also understand the nature of open source licenses within the python environment in which it could be developed and disseminated to the community.
In the revised manuscript, we have made major improvements in almost all of the sections. To address the potential issue of novelty, we have expanded the introduction section with additional paragraphs in line 128 to 168. Then, please check the revision to correct our conceptual misunderstanding about coherence and alpha in the theoretical review section. Moreover, we have made the more detailed definition of Indonesian PER result extracted from our study in line 287-318. Then, we have rerun our LDA analysis to make multiple models as recommended by the reviewer to address random seed initialization in line 421-449. To enrich the Indonesian context of the study for the reader, we have added more comprehensive characteristics of the Indonesian PER topics in line 576 to 821. Additionally, we also use pyLDAvis to understand the connection among the PER topics in Figure 6. It can supplement the description of Indonesian PER topics (RQ1) in our manuscript. We hope our effort to these improvements has made the manuscript better and more distinct from the previous work of Odden, et al (2020).
Point 2: Next, there are some issues that are technical in nature. The authors seem to have conflated two different factors in LDA analysis, alpha and coherence. These are not, in fact, the same. Alpha is a hyperparameter that controls the relative "mixedness" (or, conversely, distinctiveness) of topics learned by LDA. Low alpha values will produce topics that are very distinct, while higher alpha will produce topics that are more similar. Coherence is an external evaluation metric that evaluates how well the topics fit with the data. More about coherence can be found in M. Röder, A. Both, and A. Hinneburg, Exploring the space of topic coherence measures, in WSDM 2015 Proceedings of the Eighth ACM International Conference on Web Search and Data Mining (Association for Computing Machinery, New York, 2015), p. 399, https:// doi.org/10.1145/2684822.2685324. Because these two metrics are not the same, the authors both need to correct their explanation of them in the paper and (potentially) redo their analysis anywhere where they have conflated the two. Given the range of reported coherence scores I suspect that it is primarily a presentation issue, but that needs to be fixed
Response 2: We apologize to our misunderstanding about the coherence measure and alpha parameter. The comment is correct. We are beginners in using this methodology and your correction to this area truly influences our understanding about correct knowledge. We have revised the addressed correction in accordance with your suggestion in line 207, 371-372, 375, 406, 409-450.
Point 3: In the authors' evaluation of coherence to find topic number, they only seem to have evaluated a single model per K-value. Coherence scores can vary quite widely based on the random seed initialization of the LDA model. I suggest the authors create additional models at each K-value (around 10 is often sufficient) and evaluate their coherence scores as well to see if there is significant variation.
Response 3: Thanks very much for your constructive comment. This is truly influential for our methodological understanding about potential issues about the LDA algorithm. It must be appreciated to enrich our knowledge about the issue of random seed initialization on the LDA algorithm. We have produced Figure 5 in the revised manuscript based on multiple trained LDA models that we have rerun. We have made a new paragraph in line 414-450 to address your comment along with the fifth comment below.
Point 4: It was unclear to me whether the authors are actually analyzing Indonesian PER or whether they are analyzing a larger literature base that just happens to include some Indonesian researchers. Given their research questions I would have expected the first, but the literature they use seems to come from a much wider variety of conference proceedings including papers both outside of Physics Education Research and authors outside of Indonesia. Given their title, abstract, and research questions, I suggest the authors elaborate on how they ensured that the literature base focused on Indonesian Physics Education Research or else modify their study framing to include this wider set of disciplines and nationalities.
Response 4: In accordance with your fourth comment, the Indonesian PER literatures (as covered in our title of paper) should be defined as research articles that have been organized, peer-reviewed, and published by Indonesian PER ecosystems. It can be implied since the Abstract section in line 9 in which the term “disseminated” is clearly mentioned. Dissemination of studies must be processed by the Indonesian PER community to be published and worthy to be read. We highlighted it again in line 19-20, 272-274, 301-303. Even though the name of conferences were framed in terms of “Mathematics and Science Education”, we have carefully chosen articles published in the PER section only in this study. We have described it in line 282-293. Therefore, the context studied outside Indonesian PER themes should have been avoided.
Indonesian PER literatures can be represented through five conference proceedings that have been chosen by our study. In line 44-51 of the introduction section, we have described how these oldest five conferences have been organized by four Indonesian Teacher Education Institutions (TEIs) since 2014. Admittedly, the nature of an “international” conference is to recognize non-Indonesian authors who are able to contribute throughout the conferences. Accordingly, one can argue that these can be misinterpreted and that the selected papers do not represent the Indonesian PER community. Nonetheless, these perceptions should be invalid if we could remember that they are organized, peer-reviewed, and published by Indonesian PER scholars or even discussed and presented during the parallel session in the seminar. Moreover, the representation of authors acknowledged as Indonesian were still most of the data corpus (~ 98%). The contribution of authors from neighboring countries could not be avoided since implicitly they must influence how our discourses shape our understanding within Indonesian PER literatures. There is a possibility these articles would be discussed and cited by Indonesian PER scholars in the future study.
To address this potential issue of ambiguity, we have made a new paragraph about our definition about Indonesian PER literatures, particularly in line 298-318.
Point 5: It was also unclear to me how the authors chose the random initialization seed for their final LDA model. This was a significant focus in the Odden et al. article, since different random seed values can produce significantly different sets of topics. It is good practice for LDA analyses to clarify the procedure used to choose this parameter. If they did not use any specific procedure, that is a flaw in the methodology.
Response 5: Kindly review our response in the foregoing point 3.
Point 6: Finally, there were a few specific statements in the article that were factually wrong:
- Line 170: "Each entry in the document–word matrix represents the probability of word mentioned in each document." These entries represent the count of the words, not the relative probability
- Line 314: " The descriptor of topics is “more coherent” when alpha is more higher and near to unity [74]. " This is incorrect, as alpha and coherence score are different measures.
- Line 527: "Three rolling windows averaged the prevalence values for each year with those of the former and subsequent year. " The authors likely meant "three-year rolling windows" since this is what was done in the Odden et al. analysis.
Response 6: We have made revisions to address the second reviewer’s comments in line 219-221, 417-419, and 865-867.

Round 2
Reviewer 1 Report
I thank the authors for addressing my comments. I would suggest that they turn in a version where the changes are marked differently than version control, maybe underlined or with a line in the paragraphs that change, as the current presentation makes it hard to read the final product.
I would like to state that I believe the main contribution of this paper is not novelty, but that it can be a good example about how to apply NLP for topic analysis. This because the results in the Indonesian PER community are consistent with results obtained in other methods. Thus, it can be used as the basis for other communities, countries, or both, making it relevant and interesting for the journal. However, I also believe that this version has worse readability than before.
The discussion in section 3.1 about arguments that a reader may potentially assume its unnecessary. I thank the authors for expanding on how the data is collected and the assumptions (only journals organized, peer-reviewed and published by the PER ecosystem, represented by the five oldest that fulfill that criteria). The value to the reader is how data was collected, so this can be applied in another context. Thus, perceptions and assumptions about what the reader may think do not play a part here. I suggest a rewrite of this section focusing on how the papers were collected in case anybody wanted to reproduce this work or use it as a basis for their own country/community.
The discussion about the research question in the introduction gets sidelined by the discussion about what novelty is, and the argumentation of why this paper is novel. I think this paper should focus on the contribution (applying LDA for topic discovery in the Indonesian PER papers, and the good analysis of the results), as this could be of more interest to the reader.
The first novelty argument is that this is done in a different context. I disagree that using a known method (LDA) on different data (Indonesian vs International PER literature) is novel. But I believe it is valuable, and much more if it confirms other studies that have been done before (The references [12-15] that you mention).
The second argument is that “little is known about the LDA method to conduct topic modeling in the Indonesian PER community”. However, this is not a contribution of this paper, as the study focuses on the method, and not on how it can be better known in the community. That would require a dissemination and awareness campaign, and then measure if there is a change on how much the LDA method is known and its usefulness for the community. So, not being a direct contribution, it is also not an argument for novelty and i suggest you remove it. Also, if the focus as you say it is to “Trigger other Indonesian PER scholars to analyze the wider Indonesian PER literatures in the future” maybe you need to reconsider the scope of the journal, as a paper with an aim that local wouldn’t be interesting to the community that this journal reaches.
For the third argument, you say that you compare LDA with something you call the “traditional method”, and since it was limited you decided on LDA. This is a methodological decision, not a novelty factor. Furthermore, to be a contribution apart from the main one, this should include a discussion of the “traditional method” in your theoretical review, discussion the pros and cons. After, you would need to discuss the pros and cons of using LDA and with that comparison in mind, the reader can see in which context each method is useful. Without that comparison, this cannot be considered a contribution, nor a novelty factor of your paper.
You mention in the conclusion that “Admittedly, there are several limitations in using LDA to analyze research literatures.” Please discuss these limitations and not just mention them.
Author Response
Response to Reviewer 1 Comments
Point 1: I thank the authors for addressing my comments. I would suggest that they turn in a version where the changes are marked differently than version control, maybe underlined or with a line in the paragraphs that change, as the current presentation makes it hard to read the final product.
Response 1: The feature of track changes within Microsoft Word has been activated during the process of manuscript improvement. It might be hard to read since we conduct many revisions of almost the whole part of the manuscript based on two reviewers’ comments. We may select the “No Markup” option to display our final product without the history of improvement that we have made. In summary, we have conducted many improvements to address your valuable comments. The Introduction has been rewritten to strengthen the justification of the contribution of our paper. We have made a new paragraph to briefly discuss traditional thematic analysis in line 155-172. The data collection technique has been rewritten to address the potential issue highlighted by the first reviewers’ comment. We have extended some more paragraphs in the discussion section and the limitations of LDA have also been more explained in line 1046-1065.
Point 2: I would like to state that I believe the main contribution of this paper is not novelty, but that it can be a good example about how to apply NLP for topic analysis. This because the results in the Indonesian PER community are consistent with results obtained in other methods. Thus, it can be used as the basis for other communities, countries, or both, making it relevant and interesting for the journal. However, I also believe that this version has worse readability than before.
Response 2: The justification of our introduction has been rewritten. We hope it could make the contribution of our paper more clear, particularly for the Indonesian PER community. The implications of our work are also described in line 150-153 and 1066-1085.
Point 3: The discussion in section 3.1 about arguments that a reader may potentially assume its unnecessary. I thank the authors for expanding on how the data is collected and the assumptions (only journals organized, peer-reviewed and published by the PER ecosystem, represented by the five oldest that fulfill that criteria). The value to the reader is how data was collected, so this can be applied in another context. Thus, perceptions and assumptions about what the reader may think do not play a part here. I suggest a rewrite of this section focusing on how the papers were collected in case anybody wanted to reproduce this work or use it as a basis for their own country/community.
Response 3: The way how the data was collected has been rewritten in section 3.1, particularly in the first three paragraphs in line 274-319. We have deleted some discussions about the potential perceptions which may be emerged by the readers. They are included in the manuscript during the review process with the second reviewer to strengthen our justification to the process of the data collection. We thank the comment from the first reviewer addressing this potential issue. It must be much appreciated.
Point 4: The discussion about the research question in the introduction gets sidelined by the discussion about what novelty is, and the argumentation of why this paper is novel. I think this paper should focus on the contribution (applying LDA for topic discovery in the Indonesian PER papers, and the good analysis of the results), as this could be of more interest to the reader.
Response 4: We have rewritten the introduction section to address this comment and added the limitations of traditional thematic analysis in line 100-116 and 161-169. We hope it could make the novelty of our paper more clear to the readers.
Point 5: The first novelty argument is that this is done in a different context. I disagree that using a known method (LDA) on different data (Indonesian vs International PER literature) is novel. But I believe it is valuable, and much more if it confirms other studies that have been done before (The references [12-15] that you mention).
Response 5: We have address this comment in line 76-124
Point 6: The second argument is that “little is known about the LDA method to conduct topic modeling in the Indonesian PER community”. However, this is not a contribution of this paper, as the study focuses on the method, and not on how it can be better known in the community. That would require a dissemination and awareness campaign, and then measure if there is a change on how much the LDA method is known and its usefulness for the community. So, not being a direct contribution, it is also not an argument for novelty and i suggest you remove it. Also, if the focus as you say it is to “Trigger other Indonesian PER scholars to analyze the wider Indonesian PER literatures in the future” maybe you need to reconsider the scope of the journal, as a paper with an aim that local wouldn’t be interesting to the community that this journal reaches.
Response 6: We have deleted this argument to avoid the potential issue of outside the scope of the journal
Point 7: For the third argument, you say that you compare LDA with something you call the “traditional method”, and since it was limited you decided on LDA. This is a methodological decision, not a novelty factor. Furthermore, to be a contribution apart from the main one, this should include a discussion of the “traditional method” in your theoretical review, discussion the pros and cons. After, you would need to discuss the pros and cons of using LDA and with that comparison in mind, the reader can see in which context each method is useful. Without that comparison, this cannot be considered a contribution, nor a novelty factor of your paper.
Response 7: We have a new paragraph to discuss the brief theoretical review of traditional thematic analysis in line 155-172 and the clear advantages of the LDA model in the discussion section in line 1026-1045.
Point 8: You mention in the conclusion that “Admittedly, there are several limitations in using LDA to analyze research literatures.” Please discuss these limitations and not just mention them.
Response 8: We have made a new paragraph to discuss the potential limitations of the LDA model in line 1046-1065.
Reviewer 2 Report
This paper is much improved from the previous draft. In addition to the technical fixes I requested, I feel that the authors' extra effort to update the descriptions of the topics and provide representative references has definitely improved the presentation. The additional citations to the Odden et al. paper seem sufficient, and the additional descriptions of the text data corpus also address my previously-flagged issue.
For the most part, I feel that this article is now in a publishable state. However, there are a few small issues that I'd recommend the authors look into:
1. The labels on Figures 3-4 are very hard to read. I'm not sure what's going on there but they are very blurry.
2. For the purposes of transparency, I'd suggest the authors note the alpha value and coherence score of their final model. It was also not clear to me how they actually chose the final model (that is, the specific random seed) that corresponded with their chosen K and alpha. Did they choose this based on the coherence score? Something else? In the paper by Odden et al. that this work was based on the authors used a method in which they aggregated a large number of models, used a clustering algorithm to find the "average" topics, and then chose the closes model to this aggregate. I do not see any mention of a similar method here, so more transparency on how this model was chosen would be helpful.
3. The graph of average topic prevalence appears to have changed from The previous draft to the current draft. What caused this change? Did the authors modify their analysis or rolling window in some way?
Author Response
Response to Reviewer 2 Comments
Point 1: The labels on Figures 3-4 are very hard to read. I'm not sure what's going on there but they are very blurry.
Response 1: We have changed the font type. Hopefully, it could be more clear for now.
Point 2: For the purposes of transparency, I'd suggest the authors note the alpha value and coherence score of their final model. It was also not clear to me how they actually chose the final model (that is, the specific random seed) that corresponded with their chosen K and alpha. Did they choose this based on the coherence score? Something else? In the paper by Odden et al. that this work was based on the authors used a method in which they aggregated a large number of models, used a clustering algorithm to find the "average" topics, and then chose the closes model to this aggregate. I do not see any mention of a similar method here, so more transparency on how this model was chosen would be helpful.
Response 2: The consideration of our final model was determined using the elbow plot in Figure 5 and the subsequent description in line 432-445.
Point 3: The graph of average topic prevalence appears to have changed from The previous draft to the current draft. What caused this change? Did the authors modify their analysis or rolling window in some way?
Response 3: We thank to the careful correction from the second reviewer. It is right. We apologize for the wrong insertion of Figure 7. We have replace it with the graph as the original submission.